# Androgens predispose males to monocyte-mediated immunopathology by inducing the expression of leukocyte recruitment factor CXCL1

Julie Sellau [1], Marie Groneberg [1], Helena Fehling [1], Thorsten Thye[1], Stefan Hoenow[1], Claudia Marggraff[1], Marie Weskamm [1], Charlotte Hansen[1], Stephanie Stanelle-Bertram[2], Svenja Kuehl[1], Jill Noll[1], Vincent Wolf[1], Nahla Galal Metwally [1], Sven Hendrik Hagen[3], Christoph Dorn[4], Julia Wernecke[5], Harald Ittrich[5], Egbert Tannich [1], Thomas Jacobs[1], Iris Bruchhaus [1], Marcus Altfeld [3,6] & Hannelore Lotter [1✉]

Hepatic amebiasis, predominantly occurring in men, is a focal destruction of the liver due to the invading protozoan *Entamoeba histolytica*. Classical monocytes as well as testosterone are identified to have important functions for the development of hepatic amebiasis in mice, but a link between testosterone and monocytes has not been identified. Here we show that testosterone treatment induces proinflammatory responses in human and mouse classical monocytes. When treated with 5α-dihydrotestosterone, a strong androgen receptor ligand, human classical monocytes increase CXCL1 production in the presence of *Entamoeba histolytica* antigens. Moreover, plasma testosterone levels of individuals undergoing transgender procedure correlate positively with the TNF and CXCL1 secretion from their cultured peripheral blood mononuclear cells following lipopolysaccharide stimulation. Finally, testosterone substitution of castrated male mice increases the frequency of TNF/CXCL1-producing classical monocytes during hepatic amebiasis, supporting the hypothesis that the effects of androgens may contribute to an increased risk of developing monocyte-mediated pathologies.

[1] Department of Molecular Biology and Immunology, Bernhard Nocht Institute for Tropical Medicine, Hamburg, Germany. [2] Department Viral Zoonoses - One Health, Heinrich Pette Institute, Leibniz Institute for Experimental Virology, Hamburg, Germany. [3] Research Department Virus Immunology, Heinrich Pette Institute, Hamburg, Germany. [4] Amedes Experts, Hamburg, Germany. [5] Diagnostic and Interventional Radiology and Nuclear Medicine, University Medical Center Hamburg-Eppendorf, Hamburg, Germany. [6] Department of Immunology, University Medical Center Hamburg-Eppendorf, Hamburg, Germany. ✉email: lotter@bnitm.de

Several diseases, particularly infectious and autoimmune diseases, show a marked sex bias that can be linked to chromosomal effects or to differences in steroid hormone levels[1,2]. Generally, women show more robust immune responses to antigenic challenge than men, possibly due to proinflammatory immune responses of cells expressing estrogen receptors[3,4]. By contrast, men show increased susceptibility to infection because androgens tend to suppress adaptive immune responses[5,6].

Androgens stabilize $T_H2$ cells; however, they destabilize $T_H1$ differentiation by inhibiting IL-12 signaling[7]. However, innate immune cells, e.g., male-derived monocytes, produce larger amounts of potent proinflammatory cytokines (e.g., tumor necrosis factorα (TNF) and interleukin-1β (IL-1β)) after endotoxin stimulation than female-derived monocytes[8,9]. Production of proinflammatory cytokines is detrimental because they trigger immunopathological mechanisms at the site of infection or damage[10,11]. Indeed, a substantial group of monocyte-driven diseases, including amebiasis, leishmaniasis, and hepatocellular carcinoma, are more prominent in males (both human and mice)[12–15].

Monocytes circulating in the blood are central players during infection and inflammation[16]. Murine monocytes are characterized by expression of Ly6C and can be subdivided into classical Ly6C$^{hi}$ and non-classical Ly6C$^{lo}$ cells[17]. In humans, monocytes are categorized into three subsets based on expression of CD14 and CD16: classical CD14$^+$CD16$^-$, intermediate CD14$^+$CD16$^+$, and non-classical CD14$^-$CD16$^+$ monocytes[17–19]. These monocyte subsets in humans and mice can be characterized further according to expression of additional markers such as CCR2 and CX$_3$CR1, which are important for monocyte recruitment and adhesion to endothelial cells[18,20].

Classical monocytes form the first wave of host defense against pathogenic microorganisms but can trigger immunopathology if inadequately controlled[21,22]. C–C chemokine ligand 2 (CCL2), secreted by many cells in injured or inflamed tissue, induces egress of CCR2$^+$ classical monocytes from the bone marrow, thereby enabling migration to sites of infection[22]. Until now, it is unclear whether CCL2 guides classical monocytes directly to infected or inflamed sites in a gradient-dependent manner[23,24].

TNF induces CCL2 production by epithelial and endothelial cells[25]; thus infection correlates with high serum levels of TNF and CCL2[10,22]. Another chemokine induced by TNF, is chemokine (C-X-C motif) ligand 1 (CXCL1), also responsible for leukocyte recruitment to sites of inflammation[26,27]. CXCL1 binds to its receptor, CXCR2, and recruits neutrophils to sites of infection[26]; however, CXCR2 is also expressed by classical monocytes[28], suggesting that monocytes are also recruited via sensing of CXCL1.

Our previous study showed that recruitment of classical monocytes plays a crucial role in the mouse model of hepatic amebiasis, a parasitic disease induced by the protozoan *Entamoeba histolytica* (*E. histolytica*)[10]. The model reflects the androgen-driven sex differences observed in humans; indeed, >80% of individuals developing amebic liver abscess (ALA) are men despite higher infection rates in women[12]. So far, only CCL2 is known to be present at higher concentrations in men than in women[29]. Depleting neutrophils and monocytes from the mouse model using an αGR-1 antibody or using CCR2-knock-out, which hampers egress of Ly6C$^{hi}$ monocytes from the bone marrow[21], leads to a marked reduction in abscess formation, suggesting that Ly6C$^{hi}$ monocytes play a major role in liver damage. These results are supported by the finding that tissue destruction during ALA is mediated primarily by TNF[10].

Here, we investigate how classical monocytes from male and female individuals differ in terms of their immune responses. By analyzing humoral, cellular, transcriptomic, and hormonal responses, we provide novel insight into the increased susceptibility of men to infectious diseases due to an androgen-driven proinflammatory phenotype of classical monocytes. Especially the expression of the proinflammatory chemokine CXCL1 by classical monocytes from mice and humans is enhanced by the presence of androgens. The results support sex-based differences within the monocyte compartment, which lead to excessive recruitment of leukocytes in males, thereby increasing the risk of immune-mediated pathology.

## Results

**Amebic liver pathology is more pronounced in male mice.** The mouse model of hepatic amebiasis exhibits the same sex differences as humans, with a clear bias towards males. Ly6C$^{hi}$ monocytes and TNF are mainly responsible for immunopathology in male mice[10,30]. To gain deeper understanding of the observed sex differences, we analyzed the course of infection in mice receiving the highly pathogenic *E. histolytica* clone B2[31]. A longitudinal MRI-based analysis revealed a stable sex dimorphism between male and female mice (Fig. 1a, b), confirming previous findings using the original HM-1:IMSS isolate[30]. This difference in disease development between the sexes was reflected by higher plasma levels of TNF, CCL2, CXCL1 (Fig. 1c–e), and other cytokines (Supplementary Fig. 1a) in male mice. TNF, CCL2, and CXCL1 were mainly produced by ex vivo-restimulated liver-derived Ly6C$^{hi}$ monocytes from *E. histolytica*-infected male and female mice on Day 3 post-infection (p.i.), rather than by Ly6C$^{lo}$, Ly6C$^{neg}$, Ly6G$^+$, and CD11b$^{neg}$ cells (Fig. 1f, g). On Day 3 post-*E. histolytica* infection, TNF and CXCL1, but not CCL2, were produced by Ly6C$^{hi}$ monocytes with a clear bias towards males (Fig. 1h–j).

Since the impact of CXCL1 on abscess development had not been investigated until now, it was of crucial importance to analyze the role of this chemokine during *E. histolytica* infection. Both CCL2 and CXCL1 were upregulated rapidly upon infection of liver tissue (Supplementary Fig. 1b). To gain deeper understanding of the role of CXCL1 during hepatic amebiasis, male mice were treated with a CXCL1-depleting antibody 1 day before and 1 day after infection. On Day 3 post-*E. histolytica* infection, abscess volume (Fig. 1k) and the percentage of Ly6C$^{hi}$ cells among the total Ly6C$^+$ monocyte population in the liver (Fig. 1l) were reduced to a greater extent by the CXCL1-depleting antibody than by the isotype control.

The murine model of abscess formation due to *E. histolytica* sheds light on differences between male- and female-derived classical monocytes. To exclude the impact of the neutrophil population (Ly6G$^+$ cells) on the percentage of Ly6C$^+$ cells within the CD11b$^+$ cell population, the percentage of Ly6C$^{hi}$ monocytes was calculated in relation to the total number of Ly6C$^+$Ly6G$^-$ monocytic cells. Analysis of circulating monocytes revealed a higher percentage of Ly6C$^{hi}$ monocytes in naïve and *E. histolytica*-infected male mice than in female mice (Fig. 2a). In the liver, this sex difference was present only on Day 3 post-*E. histolytica* infection. Moreover, the percentage of Ly6C$^{hi}$ monocytes in the liver increased only in male mice after *E. histolytica* infection, not in female mice.

These results led to further characterization of murine classical monocytes (Fig. 2b–d). Besides being more dominant on Day 3 p.i., Ly6C$^{hi}$ monocytes from male mice also exhibited higher expression of the activation marker CD86, required for the induction of further adaptive immune responses, than classical monocytes from females (Fig. 2b). Furthermore, the mean fluorescence intensity (MFI) of chemokine receptor CCR2 (which is of crucial importance for egress of monocytes from the bone marrow to the circulation) on male-derived classical monocytes in blood (Fig. 2c)

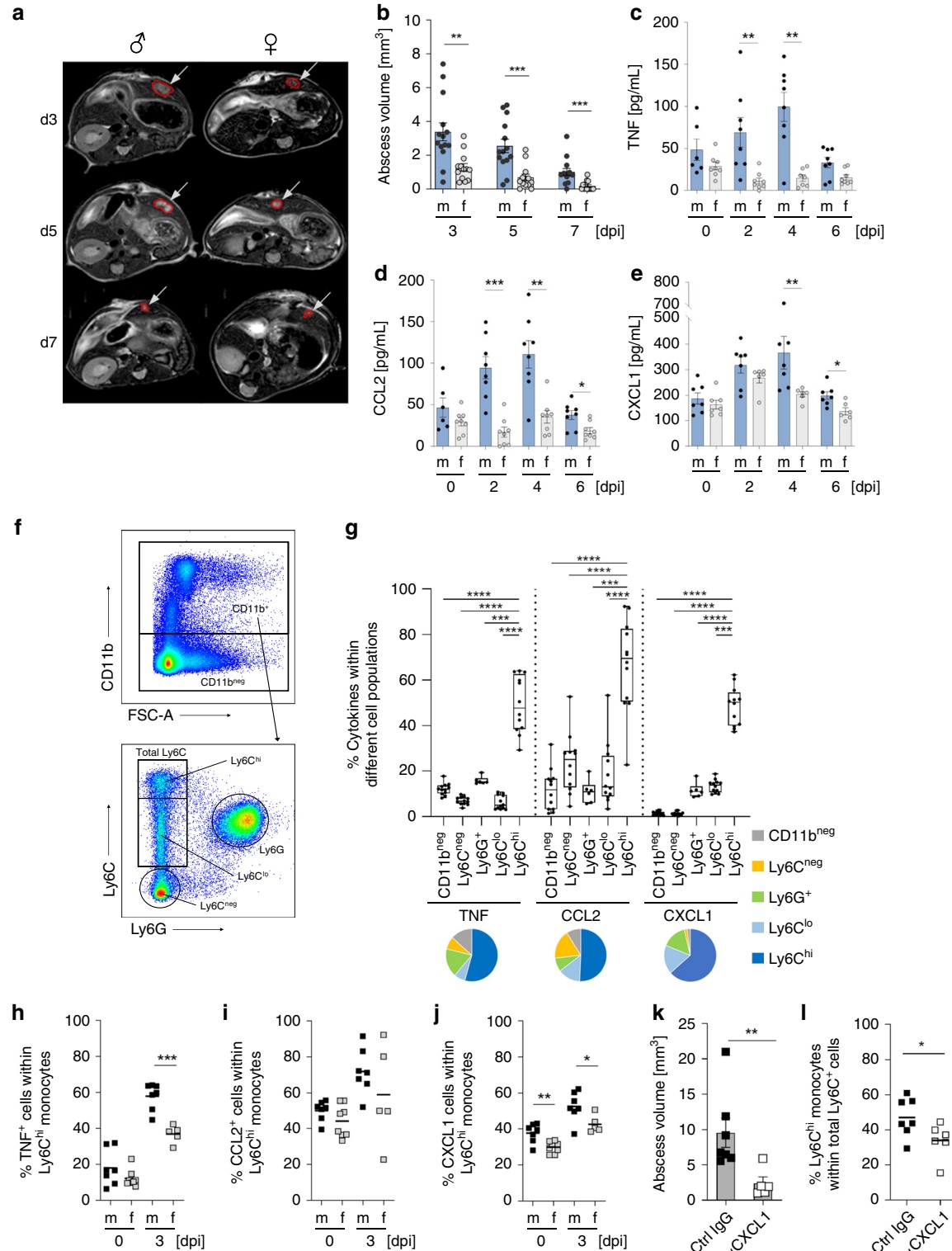

and bone marrow (Supplementary Fig. 1c) from naïve mice was higher than that on female cells[22]. In addition, expression of the chemokine receptor CX₃CR1 (important for monocyte survival and tissue homeostasis in addition to migration)[32,33] on circulating and hepatic male-derived monocytes was lower than that on female-derived monocytes on Day 3 post-*E. histolytica* infection (Fig. 2d).

Taken together, the results suggest that male-derived Ly6C$^{hi}$ monocytes have a more proinflammatory immune profile than female-derived monocytes during *E. histolytica* infection, and that they have greater potential to sense CCL2, which is essential for egress from the bone marrow to the circulation.

**Comparison of murine and human classical monocytes.** Given the functional similarities between human and murine monocytes, we asked whether our findings in murine classical monocytes were replicated by human blood-derived monocytes. In

**Fig. 1 Cytokine expression by Ly6C$^{hi}$ monocytes during murine hepatic amebiasis. a** Representative T2-weighted MRI images of male and female mouse liver tissues infected with *E. histolytica* on Days 3, 5, and 7 p.i. **b** Abscess volume in male and female mice during the course of infection ($n_f = 13$, $n_m = 14$). Plasma concentration of **c** TNF, **d** CCL2, and **e** CXCL1 in male and female mice during the course of disease (**c** $n_{m/f\ d0} = 6/8$; $n_{m/f\ d2} = 8/8$; $n_{m/f\ d4} = 8/7$; $n_{m/f\ d6} = 8/8$; **d** $n_{m/f\ d0} = 6/8$; $n_{m/f\ d2} = 8/8$; $n_{m/f\ d4} = 8/8$; $n_{m/f\ d6} = 8/8$; **e** $n_{m/f\ d0} = 7/7$; $n_{m/f\ d2} = 7/6$; $n_{m/f\ d4} = 7/6$; $n_{m/f\ d6} = 7/6$; pooled data from two independent experiments; Depicted are the means ± SEM). **f** Representative gating strategy used to identify murine CD11b$^{neg}$, Ly6C$^{neg}$, Ly6G$^+$, Ly6C$^{lo}$, and Ly6C$^{hi}$ subpopulations by flow cytometry. **g** Intracellular expression of TNF, CCL2, and CXCL1 by murine liver-derived cell subpopulations identified in **f** on Day 3 p.i. ($n_{Ly6Chi,\ Ly6Clo,\ Ly6Cneg,\ CD11bneg} = 12$; $n_{Ly6G+} = 6$, male and female mice; pooled data of two independent experiments; Depicted are box ranging from 25th to 75th percentile and whiskers from min to max including the median). Intracellular staining of **h** TNF, **i** CCL2, and **j** CXCL1 in liver-derived Ly6C$^{hi}$ monocytes from naïve and *E. histolytica*-infected male and female mice on Day 3 p.i. (**h** $n_{m/f\ d0} = 7/8$; $n_{m/f\ d3} = 7/5$; **i** $n_{m/f\ d0} = 7/8$; $n_{m/f\ d3} = 7/5$; **j** $n_{m/f\ d0} = 7/8$; $n_{m/f\ d3} = 7/5$; pooled data of two independent experiments; Depicted are the means). **k** Abscess volume in αCXCL1-treated and *E. histolytica*-infected male mice on Day 3 p.i. ($n_{\alpha CXCL1} = 5$; $n_{ctrl\ IgG} = 7$; pooled data of two independent experiments; Depicted are the means ± SEM). **l** Graph showing the percentage of Ly6C$^{hi}$ monocytes within the total liver-derived Ly6C$^+$ monocyte population within CD11b$^+$ cells from αCXCL1-treated and *E. histolytica*-infected male mice on Day 3 p.i. ($n_{\alpha CXCL1} = 5$; $n_{ctrl\ IgG} = 7$; pooled data of two independent experiments; depicted are the means). *P*-values were calculated using two-tailed grouped analysis: Student's *t*-test (**h**, **j**, **l**) or the Mann–Whitney test (**b–e**, **g**, **k**), *$P < 0.05$; **$P < 0.01$; ***$P < 0.001$; ****$P < 0.0001$). Source data are provided as a Source Data file.

contrast to mice, where males have a higher percentage of classical Ly6C$^{hi}$ cells than female mice (Fig. 2a), we found no sex-based differences between healthy volunteers with respect to distribution of the three different monocyte subsets (classical (CD14$^+$CD16$^-$), intermediate (CD14$^+$CD16$^+$), and non-classical (CD14$^-$CD16$^+$)) in the blood (Fig. 3a, b). Similar to classical monocytes from naïve mice (Fig. 2b), there was no difference in expression of CD86 and CX$_3$CR1 by peripheral classical monocytes from healthy men and women (Fig. 3c, e). However, the observed sex difference in CCR2 expression by circulating murine classical Ly6C$^{hi}$ monocytes was similar in humans: male-derived classical monocytes showed more pronounced expression of CCR2 compared to females (Fig. 3d).

To further analyze whether sex-based differences between activated human monocytes are present at a transcriptional level, we performed RNA sequencing analysis of LPS-stimulated classical monocytes. For this purpose, we isolated PBMCs from women and men, stimulated them with LPS, and separated CD14$^+$ monocytes for RNA isolation and cDNA library preparation. Overall, transcripts showing a log2-FoldChange>2 and a padj value<0.05 in LPS-treated samples compared with medium control samples were included in the gene comparison analysis. Overall, 1232 gene transcripts were expressed at similar levels by monocytes from men and women; however, 439 genes were expressed by male-derived monocytes, and 531 were expressed by female-derived monocytes (Fig. 3f). These three groups of genes were used for further PANTHER GO-slim term analysis (Fig. 3g)[34]. Interestingly, only a small fraction of genes was involved in the GO term "immune system processes" (GO:0002376), independent of sex. The genes in GO:0002376 were further sub-grouped into three more terms: "leukocyte migration" (GO:0050900), "immune effector migration" (GO:0002252), and "immune response" (GO:0006955). The term "leukocyte migration" included more male-derived transcripts (55.6%) than female-derived transcripts (41.2%). Since GO:0050900 included more genes relevant to leukocyte migration, the assumption was that GO terms including the tags "migration", "chemotaxis", and "motility" would be more common among male-derived transcripts. Indeed, a more detailed GOrilla analysis of these GO terms (Table 1) showed significantly lower FDR-$q$ values for the male-derived transcriptome, indicating a stronger gene expression profile for migration (Fig. 3h). However, in contrast to the overall more pronounced expression pattern for migration in male-derived classical monocytes upon LPS stimulation, the numbers of significantly upregulated transcripts related to the GO term "cytokine production" (GO:0001816) was generally higher in female- than in male-derived monocytes (112 and 95, respectively; Supplementary Fig. 2a). *TNF* transcripts were the

exception; *TNF* was among the most significantly differentially upregulated genes in the male-derived monocyte subset. Interestingly, the expression of genes with immune regulatory functions was more abundant in female- than in male-derived monocytes, with *EXOSC6* being the most prominent candidate gene (padj = 1.75E-05) playing a role in mRNA degradation[35]. These data suggest higher counter-regulatory potential in female monocytes (Supplementary Fig. 2b, Supplementary Table 1).

Taken together, human and murine classical monocytes show similar sex-related differences: equal expression of CD86 and CX$_3$CR1, and higher expression of CCR2 by male-derived monocytes. Furthermore, classical monocytes from men show a stronger transcriptomic profile for migration, chemotaxis, and motility than those from women under LPS stimulation.

**Cytokine expression of murine and human classical monocytes.** To identify putative parallels in the sex bias of cytokine expression patterns by classical monocytes from mice infected with *E. histolytica* and humans, we stimulated PBMCs from healthy men and women ex vivo with LPS and an *E. histolytica*-derived antigen preparation comprising the TLR4-stimulant lipopeptidophosphoglycan[36,37]. We then examined TNF, CCL2, and CXCL1 production by classical CD14$^+$CD16$^-$ monocytes by flow cytometry. Most prominently, LPS stimulation led to significantly stronger expression of CXCL1 by classical monocytes from men (Fig. 4d), similar to the result observed in restimulated classical monocytes from *E. histolytica* infected male mice (Fig. 1j). To gain deeper insight into the expression patterns of TNF, CCL2, and CXCL1 by human classical monocytes, we concatenated classical monocytes from the previous experiment by sex and then analyzed them in parallel using Hierarchical Stochastic Neighbor Embedding (HSNE)[38,39] (Fig. 4e, f). We identified three clusters showing differential expression of TNF, CXCL1, and CCL2 in both sexes after stimulation by *E. histolytica* or LPS. For each cluster, we analyzed the ratio of the percentage of male- and female-derived monocytes. Independent of the stimulus, female-derived classical monocytes were more abundant in the cluster of TNF$^+$/CXCL1$^-$/CCL2$^-$ (Cluster 1: M:F; *E. his* [%] 38:62; LPS [%] 36:64). The cluster that included TNF$^+$/CXCL1$^+$/CCL2$^+$-expressing cells contained more male-derived monocytes (Cluster 3: M:F; *E. his* [%] 69:31; LPS [%] 65:35), independent of the stimulus (Table 2).

To summarize, as in mice, classical monocytes from men have the potential to produce more CXCL1 upon stimulation than monocytes from women. Additionally, they produce more of the three cytokines of interest (TNF, CCL2, and CXCL1) simultaneously, independent of the stimulus used.

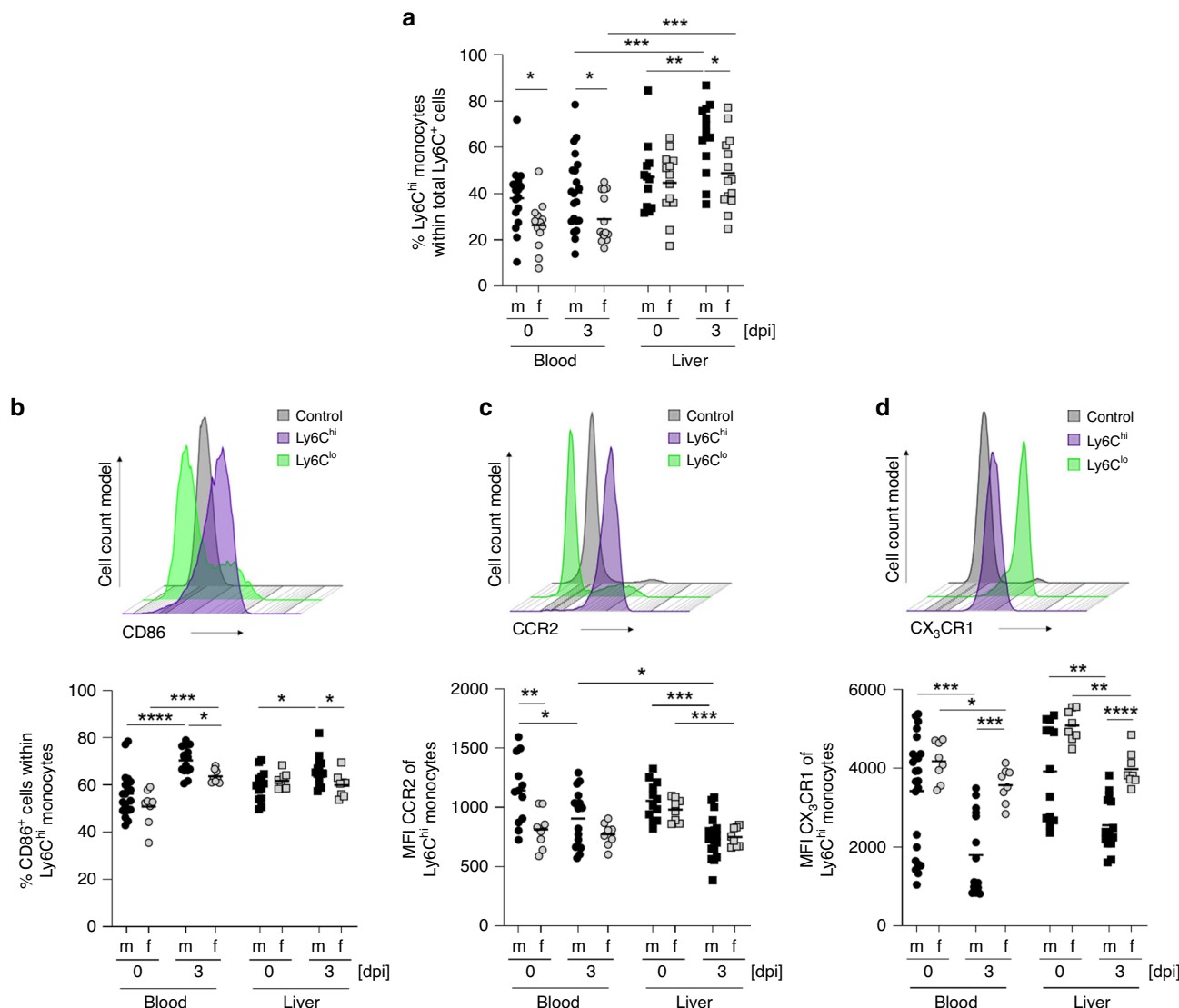

**Fig. 2 Male mice show higher percentages of classical monocytes upon infection. a** Percentage of Ly6C$^{hi}$ monocytes within the total Ly6C$^+$ monocyte population of CD11b$^+$ cells isolated from the blood and liver of naïve and *E. histolytica* (Day 3 p.i.)-infected mice (blood: $n_{m/f\ d0} = 16/13$; $n_{m/f\ d3} = 21/14$; liver: $n_{m/f\ d0} = 12/13$; $n_{m/f\ d3} = 13/14$; pooled data of four independent experiments; depicted are the means). Histogram and percentage of **b** CD86$^+$ cells, and the **c** MFI of CCR2 and **d** CX$_3$CR1 of Ly6C$^{hi}$ monocytes in the blood and the liver of naïve and *E. histolytica*-infected mice (Day 3 p.i.), as determined by flow cytometry (**b** blood: $n_{m/f\ d0} = 19/8$; $n_{m/f\ d3} = 15/8$; liver: $n_{m/f\ d0} = 13/8$; $n_{m/f\ d3} = 15/8$; **c** blood: $n_{m/f\ d0} = 12/8$; $n_{m/f\ d3} = 17/8$; liver: $n_{m/f\ d0} = 12/8$; $n_{m/f\ d3} = 17/8$; **d** blood: $n_{m/f\ d0} = 24/8$; $n_{m/f\ d3} = 15/8$; liver: $n_{m/f\ d0} = 13/8$; $n_{m/f\ d3} = 15/8$; pooled data of two independent experiments; depicted are the means). *P*-values were calculated using two-tailed grouped analysis Mann–Whitney test (**a–d**), *$P < 0.05$; **$P < 0.01$; ***$P < 0.001$; ****$P < 0.0001$). Source data are provided as a Source Data file.

**Autocrine TNF enhances CXCL1 but not CCL2 expression.** Besides regulating apoptosis and cell differentiation, TNF modulates secretion of different cytokines, i.e., CCL2 and CXCL1, shown in epithelial and endothelial cells[25,27]. Thus, we asked whether TNF induced by LPS has an autocrine effect on expression of CXCL1 or CCL2 by classical monocytes. To test this, TNF was depleted from the supernatant of LPS-stimulated classical monocytes from men using a specific antibody. The results revealed impaired secretion of CXCL1 (Fig. 5a) but not CCL2 (Fig. 5d), indicating that TNF has a putative autocrine effect on CXCL1 production by monocytes. TNF signals via two receptors, TNFRI and TNFRII. Blocking TNFRI or TNFRII, or both, using specific monoclonal antibodies markedly reduced production of CXCL1 (Fig. 5b), but not CCL2 (Fig. 5e). Again this suggests that CCL2 production by classical human monocytes is independent of TNFR engagement after LPS stimulation.

However, we found no sex-specific differences in expression of the two TNF receptors by classical monocytes (Supplementary Fig. 3c, d).

Next, we asked whether production of CXCL1 and CCL2 is mediated by classical signaling pathways, including p38 MAPK and PI3K, or by transcription factors AP-1 and NFκB. To do this, we used standard inhibitors, including actinomycin D (an inhibitor of DNA transcription), as a positive control for inhibition. Production of CXCL1, and particularly CCL2, was inhibited markedly by blocking these checkpoint molecules (Fig. 5c, f). In contrast, inhibiting JNK led to a marked increase in CXCL1 secretion by LPS-stimulated classical monocytes (Fig. 5c); CCL2 levels were unaffected (Fig. 5f). Based on the study by Lo et al.[27], Cai et al.[40], and Koryankina et al.[41], who examined CXCL1 expression in human endothelial cells and the interaction with androgens, respectively[27,40,41], we propose a

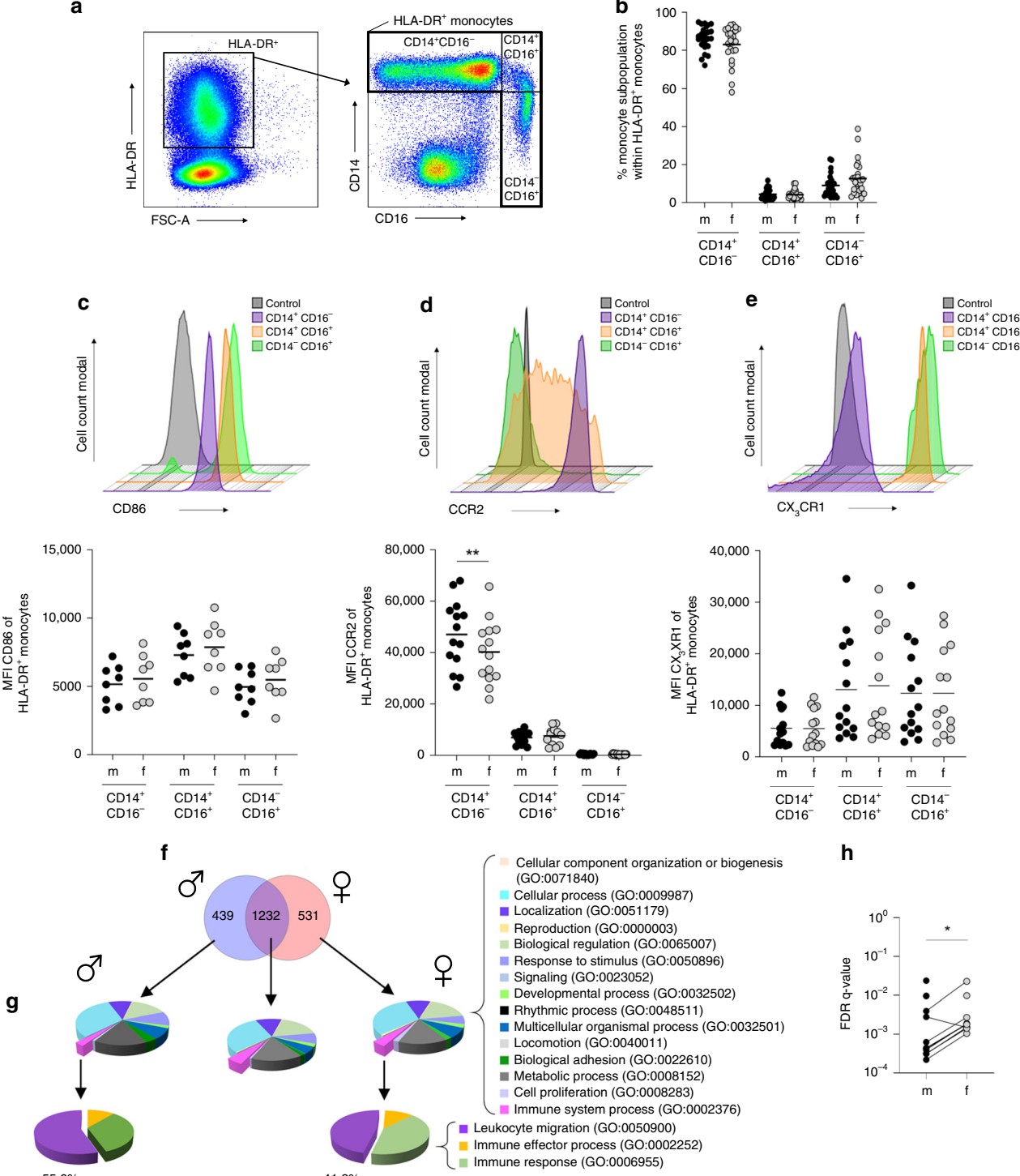

**Fig. 3 Monocytes from healthy men show stronger expression profile for migration. a** Gating strategy to identify monocyte subsets in healthy human blood donors. **b** Percentage of classical (CD14+CD16−), intermediate (CD14+CD16+), and non-classical (CD14−CD16+) monocytes in the peripheral blood of healthy men and women ($n_{m/f\ CD14+CD16-} = 24/25$; $n_{m/f\ CD14+CD16+} = 24/25$; $n_{m/f\ CD14-CD16+} = 24/25$; pooled data from samples collected over a time frame of nine months; depicted are the means). Histogram and graphs showing the MFI of **c** CD86, **d** CCR2, and **e** CX$_3$CR1 on peripheral monocyte subpopulations from healthy male and female human blood donors (**c** $n_{m/f\ CD14+CD16-} = 8$; $n_{m/f\ CD14+CD16+} = 8$; $n_{m/f\ CD14-CD16+} = 8$; **d** $n_{m/f\ CD14+CD16-} = 14$; $n_{m/f\ CD14+CD16+} = 14$; $n_{m/f\ CD14-CD16+} = 14$; **e** $n_{m/f\ CD14+CD16-} = 14$; $n_{m/f\ CD14+CD16+} = 14$; $n_{m/f\ CD14-CD16+} = 14$; pooled data of samples collected over a time frame of nine months; Depicted are the means). **f** Transcriptome analysis of CD14+ monocytes from men and women. PBMCs from male ($n = 4$) and female ($n = 3$) healthy donors (25–49 years of age) were stimulated with LPS, and CD14+ monocytes were isolated using MACS. RNA was isolated, transcribed into cDNA, and subjected to RNA Seq analysis. Venn diagram showing differentially regulated genes from men and women following LPS stimulation compared with the corresponding medium control (log2FoldChange > 2 and a padj value < 0.05). **g** PANTHER GO-slim analysis of biological processes, and **h** FDR-$q$ values of GO terms connected to leukocyte migration in men and women conducted with GOrilla analysis(Table 1). *P*-values were calculated using two-tailed paired analysis; Wilcoxon matched-paired signed rank test (**d**, **h**), *$P < 0.05$; **$P < 0.01$. Source data are provided as a Source Data file.

**Table 1 GO terms connected with migration, chemotaxis, and motility.**

| GO term | Description | FDR q-value m | FDR q-value f |
|---|---|---|---|
| GO:0097529 | Myeloid leukocyte migration | 2.21E-04 | 1.05E-03 |
| GO:0030595 | Leukocyte chemotaxis | 3.12E-04 | 1.52E-03 |
| GO:0030593 | Neutrophil chemotaxis | 3.21E-04 | 1.42E-03 |
| GO:0060326 | Cell chemotaxis | 4.01E-04 | 1.88E-03 |
| GO:1990266 | Neutrophil migration | 4.33E-04 | 1.80E-03 |
| GO:0071621 | Granulocyte chemotaxis | 4.45E-04 | 1.84E-03 |
| GO:0097530 | Granulocyte migration | 6.30E-04 | 2.73E-03 |
| GO:0050900 | Leukocyte migration | 2.80E-03 | 1.50E-03 |
| GO:0006935 | Chemotaxis | 3.97E-03 | 2.28E-02 |
| GO:2000147 | Positive regulation of cell motility | 9.54E-03 | ns |
| GO:0030335 | Positive regulation of cell migration | 9.65E-03 | ns |
| GO:0045073 | Regulation of chemokine biosynthetic process | 2.37E-02 | ns |
| GO:0016477 | Cell migration | ns | 9.80E-03 |

scheme for monocytes (see Fig. 5g) including the putative influence of this hormone.

In summary, these results reveal differential regulation of CCL2 and CXCL1 in classical monocytes from healthy male blood donors upon LPS stimulation: autocrine action of TNF induces CXCL1, but not CCL2 production via the p38 MAPK signaling pathway.

**Effect of testosterone on TNF, CCL2, and CXCL1 release**. The effect of hormone treatment, particularly testosterone, on the immune response of individuals transitioning from one sex to another is poorly understood. Here, we examined a cohort of women transitioning to men through systemic testosterone treatment. PBMCs were collected (over a time period of 201–219 days) and stimulated for 17 h with LPS before cytokine levels in the supernatant were analyzed throughout the period of testosterone treatment. We found that testosterone had a significant effect of TNF and CXCL1 release (Fig. 6a, c), as well as on other cytokines (Supplementary Fig. 4a, c), but not on CCL2 (Fig. 6b). In parallel, increasing plasma levels of testosterone and 5α-dihydrotestosterone (DHT) during treatment correlated significantly with increased levels of TNF and CXCL1, but not with those of CCL2 (Fig. 6d–f).

Since we did not have the opportunity to identify the source of these cytokines in this approach, we treated isolated monocytes from healthy male and female blood donors with DHT and chose a moderate, instead of a strong, TLR4 stimulator. For this, we used the E. histolytica antigen preparation (E. his). In this case, DHT rather than testosterone was added to ensure direct effects on the androgen receptor (AR) in vitro; this is because aromatases can convert testosterone to estradiol, thereby facilitating signaling via the estradiol receptor as well. Before starting the experiment, we observed no differential expression of the AR between monocytes from men and women by flow cytometry (Supplementary Fig. 3b). However, we found only a slight increase in TNF levels in supernatant from male and female monocytes treated with DHT and E. his (Fig. 6g). CCL2 expression was slightly higher in male-derived monocytes, but this was independent of DHT (Fig. 6h). Furthermore, CXCL1 showed an initial and significant sex bias, with higher production by

male- than female-derived monocytes upon antigen stimulation. Moreover, DHT had an additional significant effect on release of CXCL1 by both male-and female-derived monocytes (Fig. 6i). The sex bias in monocytic cytokine production was much less clear for other cytokines (Supplementary Fig. 4g–i).

To summarize, we found that testosterone and DHT levels correlate with increased expression of CXCL1 and TNF by PBMCs from individuals undergoing transgender procedure; however, DHT has a direct effect on expression of CXCL1 (but not TNF or CCL2) by human classical monocytes.

**Testosterone treatment modulates classical monocytes**. We used the murine model of E. histolytica infection to further examine the effects of androgens in humans and to analyze their systemic effects on immune responses by monocytes. As reported previously, gonadectomy and testosterone substitution have a direct effect on the outcome of liver pathology during E. histolytica infection. Gonadectomy of male mice reduces the severity of lesions, while testosterone treatment of female mice increases abscess size at 7 days p.i. (compared with the respective control groups)[42]. To further investigate the role of androgens on early, innate immune-mediated pathology driven by classical monocytes, susceptible male mice were gonadectomized 4 weeks prior to infection and supplemented with testosterone or carrier solution after 2 weeks. After another 2 weeks, mice were infected intrahepatically with E. histolytica (Fig. 7a). Hepatic immune cells were isolated from E. histolytica-infected gonadectomized (G) or gonadectomized and testosterone-treated (T) mice, and restimulated with PMA/ionomycin. Testosterone treatment of gonadectomized mice resulted in significantly higher intracellular expression of TNF, CCL2, and CXCL1 by classical monocytes upon ex vivo restimulation when compared with the corresponding control groups (Fig. 7b). Moreover, testosterone treatment not only impacted cytokine production by classical monocytes, but also increased the frequency of Ly6C^hi monocytes in the liver during infection (Fig. 7c). Interestingly, circulating and hepatic Ly6C^hi monocytes from gonadectomized male mice, which received the testosterone treatment showed higher expression of the costimulatory receptor CD86, independent of infection and/or organ (Fig. 7d). As shown in male mice (Fig. 2c), CCR2 expression by circulating and hepatic classical monocytes fell significantly upon E. histolytica infection. Independent of infection, CCR2 expression by Ly6C^hi monocytes in the blood and liver of gonadectomized and testosterone-supplemented male mice was lower than that in the gonadectomized control group (Fig. 7e). Additionally, expression of CX₃CR1 by Ly6C^hi monocytes in the blood was lower in gonadectomized and testosterone-supplemented male mice than in gonadectomized control mice on Day 3 p.i. (Fig. 7f), reflecting the results obtained in male and female mice during infection (Fig. 2d).

Expression of CCR2 correlates negatively with upregulation of the intracellular AR[43,44]. Therefore, we compared AR mRNA expression in PBMCs from sham-operated and gonadectomized naïve male mice (Fig. 7g), and from gonadectomized and testosterone-substituted male mice, upon infection (Fig. 7h). Gonadectomy increased AR mRNA expression by PBMCs from naïve male mice, whereas testosterone treatment reduced AR mRNA expression during infection (compared to that in gonadectomized mice).

In conclusion, systemic testosterone treatment drove the innate immune response toward a more proinflammatory profile, supporting findings from the human transgender cohort. Moreover, gonadectomy and testosterone substitution in male mice replicated the original sex differences observed in the murine

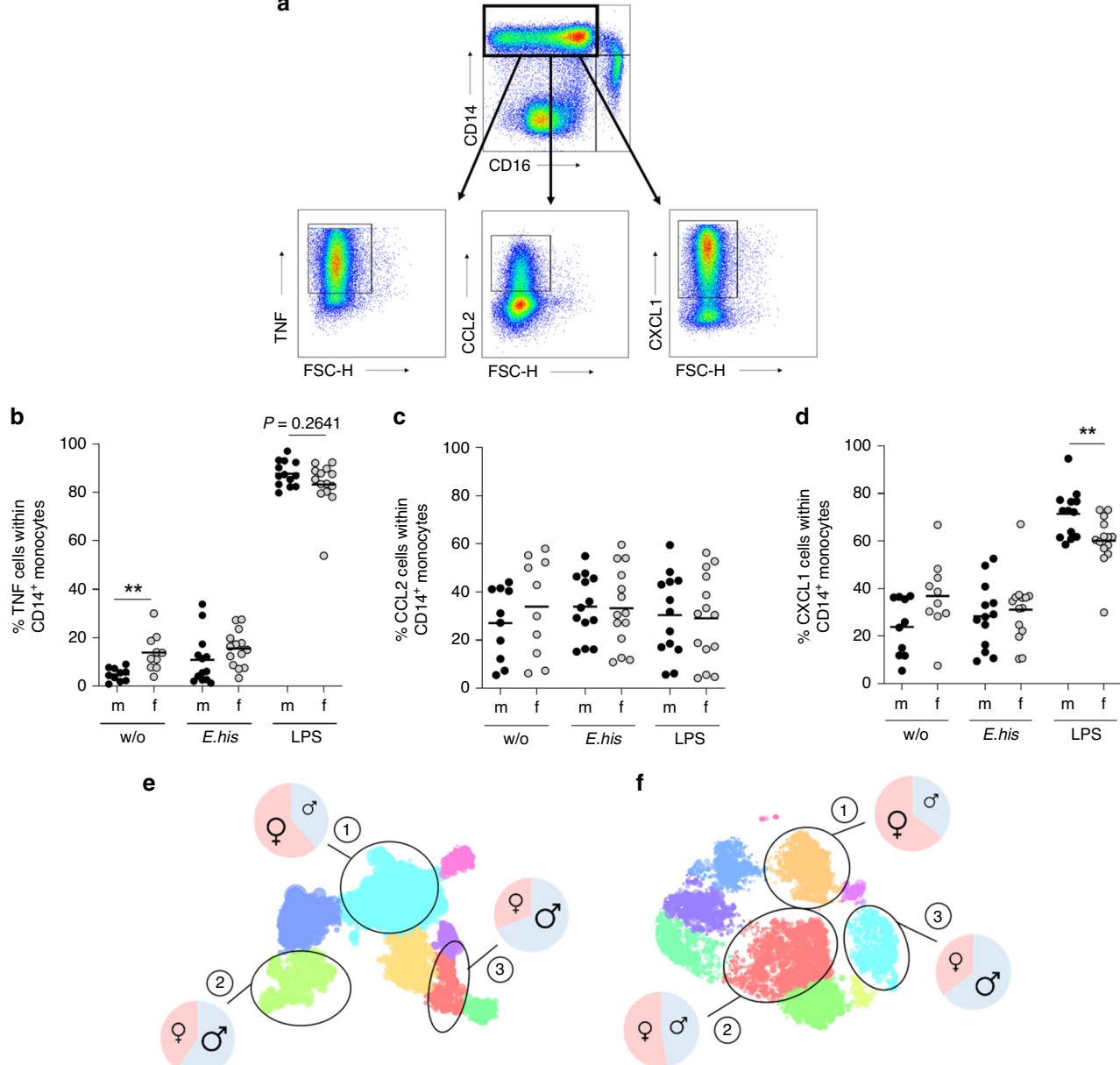

**Fig. 4 TNF+, CXCL1+, and CCL2+ classical monocytes are more abundant in men. a** Gating strategy and representative dot plots showing cytokines produced by classical monocytes upon stimulation. PBMCs from men and women were stimulated for 6 h with *E. histolytica* lysate (0.1 mg/mL) or LPS (0.1 µg/mL), and intracellular expression of **b** TNF, **c** CCL2, and **d** CXCL1 by classical monocytes was measured by flow cytometry and compared to unstimulated controls (**b** $n_{m/f\ w/o} = 10/10$; $n_{m/f\ E.\ his} = 13/14$; $n_{m/f\ LPS} = 13/14$; **c** $n_{m/f\ w/o} = 10/10$; $n_{m/f\ E.\ his} = 13/14$; and $n_{m/f\ LPS} = 13/14$; **d** $n_{m/f\ w/o} = 10/10$; $n_{m/f\ E.\ his} = 13/14$; $n_{m/f\ LPS} = 13/14$; pooled data from samples collected over a time frame of nine months; depicted are the means). Hierarchical Stochastic Neighbor Embedding (HSNE) analysis of **e** *E. histolytica*- CD14+CD16− and **f** LPS-stimulated CD14+CD16− classical monocytes. Cluster (1), TNF+CXCL1−CCL2−; cluster (2), TNF+CXCL1+CCL2−; and cluster (3), TNF+CXCL1+CCL2+ (calculated from classical CD14+ monocytes) (Table 2). *P*-values were calculated using two-tailed grouped analysis: Mann–Whitney test (**b**, **d**), \*\**P* < 0.01. Source data are provided as a Source Data file.

## Table 2 Cytokine distribution of HSNE clusters.

| Cluster | Stimulus | | | |
|---|---|---|---|---|
| | *E. histolytica* lysate | | LPS | |
| | *m* (%) | *f* (%) | *m* (%) | *f* (%) |
| 1) TNF+ CXCL1− CCL2− | 38 | 62 | 36 | 64 |
| 2) TNF+ CXCL1+ CCL2− | 59 | 41 | 47 | 53 |
| 3) TNF+ CXCL1+ CCL2+ | 69 | 31 | 65 | 35 |

model of amebiasis with respect to cytokine expression, the percentage of classical monocytes, and surface marker expression of classical monocytes except that of CCR2.

## Discussion

Male-biased diseases can be linked to the proinflammatory cytokines TNF, CXCL1, and CCL2[9,45,46]. In this study, we observed increased levels of these cytokines in plasma from male mice infected with *E. histolytica*. On Day 3 p.i., the main source of these cytokines in the immune cell compartment of *E. histolytica*-

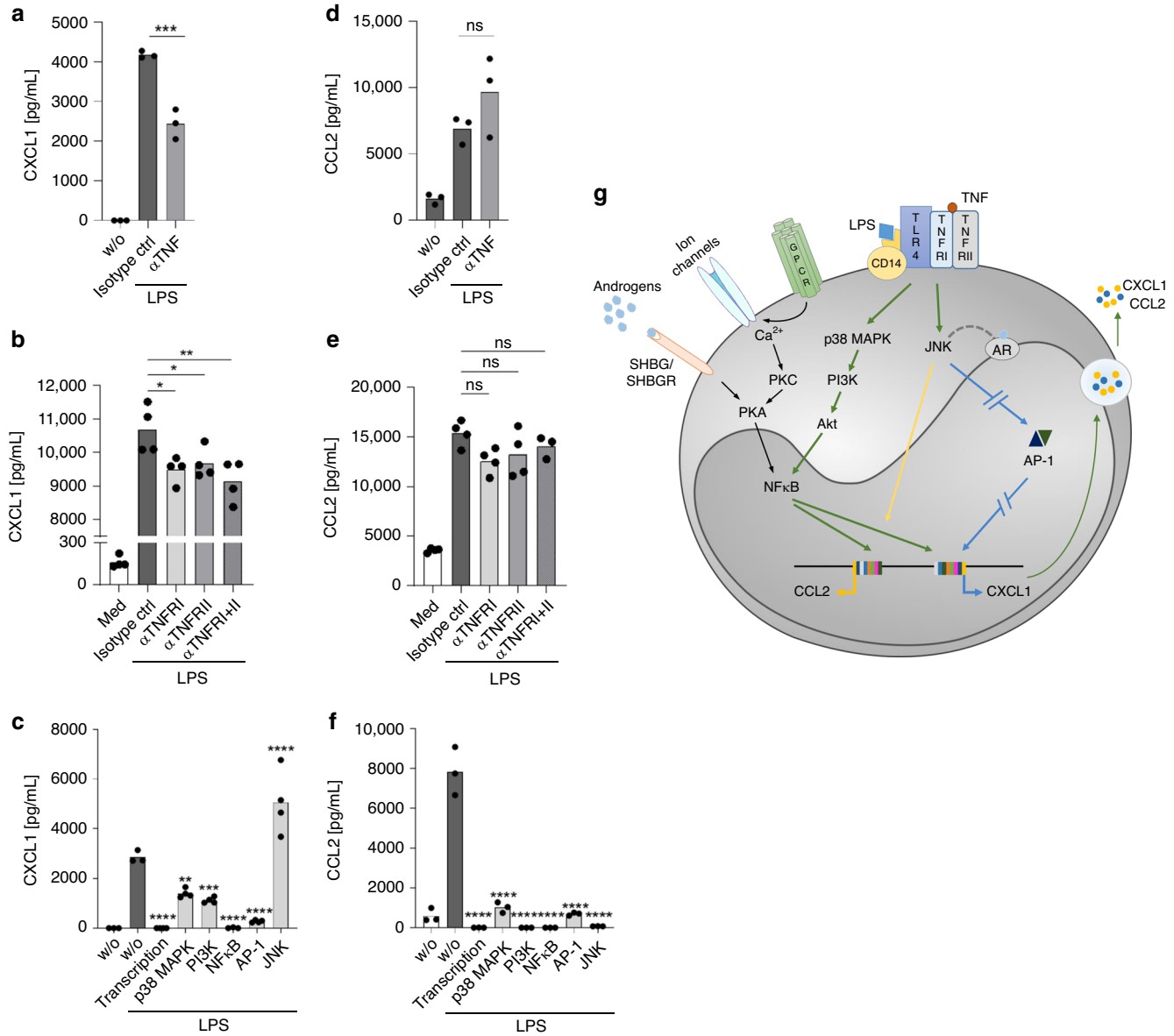

**Fig. 5 Role of TNF, the MAPK pathway, and JNK in expression of CXCL1 and CCL2.** MACS-purified CD14+ monocytes from men were stimulated for 24 h with LPS (0.1 μg/mL). The concentration of **a–c** CXCL1 and **d–f** CCL2 in the supernatant was examined as follows: **a**, **d** in the presence of an αTNF mAb or an isotype IgG mAb control ($n_{w/o; Isotype ctrl; αTNF}$ = 3); **b**, **e** in the presence of αTNFRI (1 μg/mL), an αTNFRII mAb (10 μg/mL), or an isotype control IgG mAb ($n_{w/o; Isotype ctrl; αTNFRI; αTNFRII; αTNFRI+II}$ = 4); **c**, **f** in the presence of inhibitors of DNA transcription (4 μM actinomycin D), p38 MAPK (10 μM SB202190), PI3K (10 μM LY294002), NF-κB (5 μM BAY11-7085), of AP-1 (17 μM Tanshinone) and of JNK (10 μM SP600125) (**c** $n_{w/o stimulation; w/o; Transcription; p38 MAPK; PI3K; NFκB; AP-1; JNK}$ = 3; **f** $n_{w/o stimulation; w/o; Transcription; p38 MAPK; PI3K; NFκB; AP-1; JNK}$ = 3). The data shown are one representative out of three independent experiments. **a–f** Depicted are the means. **g** Proposed scheme outlining differences in the CCL2 and CXCL1 expression by LPS-stimulated CD14+ monocytes, based on the study by Lo et al.[27], Cai et al.[40] and Koryankina et al.[41] (green line = part of the signal transduction for CCL2 and CXCL1, blue interrupted line=inhibition of CXCL1 expression, yellow line=induces CCL2, black line = induced by androgens, and gray dotted line = interaction assumed). P-values were calculated using two-tailed grouped analysis using an ordinary one-way ANOVA with Bonferoni Correction (**a–f**), *$P < 0.05$; **$P < 0.01$; ***$P < 0.001$; ****$P < 0.0001$; (**a** F (degree of freedom (DFd) (12, 6)) = 273, 9; **b** F (4, 15) = 300, 3; **c** F (7, 21) = 48, 39; **d** F (2, 6) = 14, 10; **e** F (4, 14) = 40, 86; **f** F (7, 16) = 93,23). Source data are provided as a Source Data file.

infected livers were Ly6C[hi] monocytes, which showed a significant male bias with respect to TNF and CXCL1. Furthermore, the frequency of Ly6C[hi] monocytes in the circulation and liver was higher in males.

However, the male bias in plasma-CCL2 levels might be due to other cell types or mechanisms[47]. Similar to CCL2, TNF plays an immunopathological role during development of ALA induced by E. histolytica infection[10]. TNF is a major immune activator in response to infection; however, although we know little about its

differential expression in male and female mice[48], no report has examined the sex-biased production by Ly6C[hi] monocytes. Here, we show for the first time that male-derived Ly6C[hi] monocytes secrete more TNF than female-derived monocytes upon infection with E. histolytica, which may be one factor underlying the male bias for development of ALA[10]. We noted a similar sex bias in production of CXCL1 by Ly6C[hi] monocytes. Binding of CXCL1 to CXCR2 on neutrophils and monocytes increases recruitment[49] and is involved in several immune-mediated pathologies[50–52].

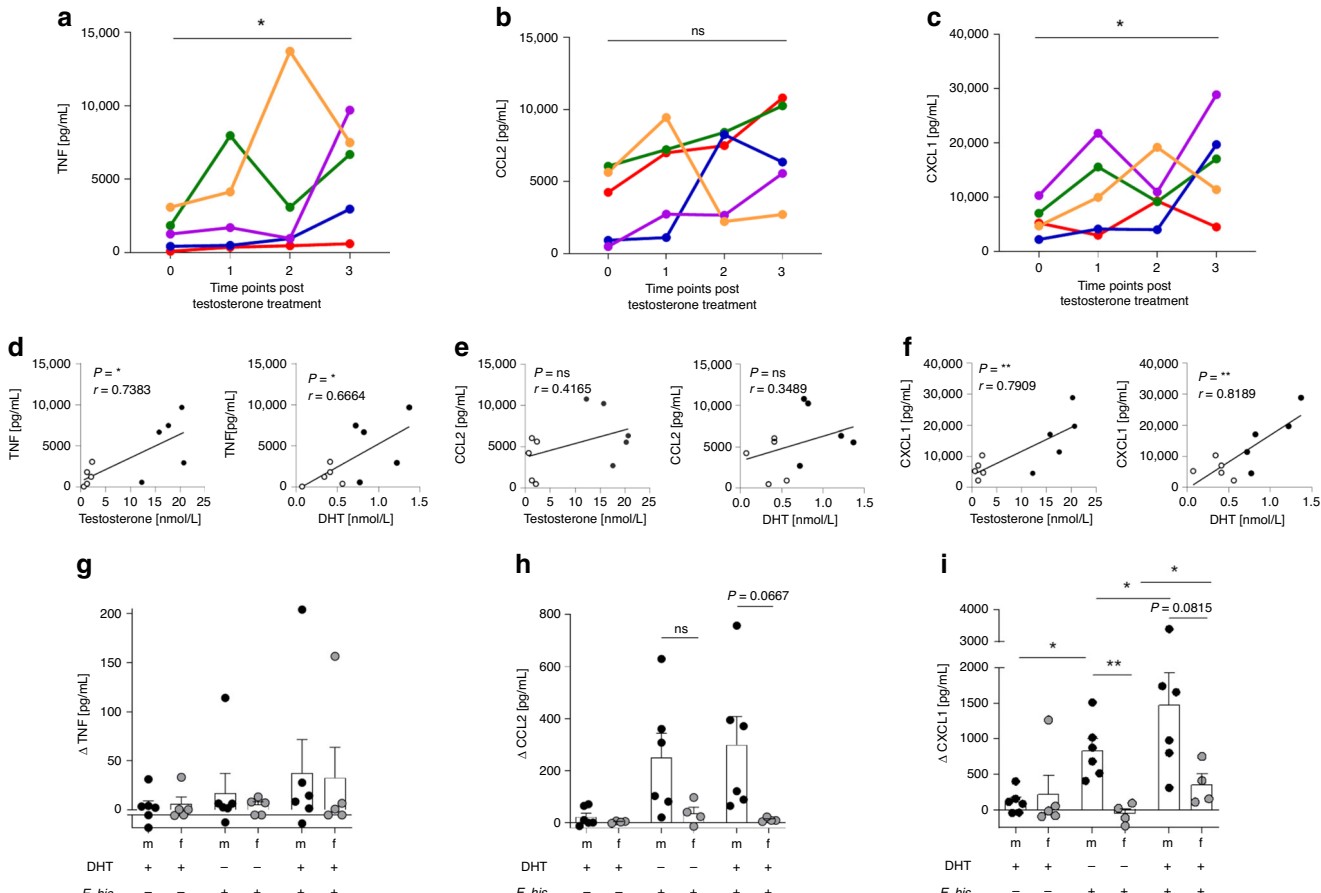

**Fig. 6 Androgens modulates secretion of TNF and CXCL1 by PBMCs from transgender men.** Cytokine production by LPS-stimulated PBMCs isolated from five women transitioning to men under testosterone treatment. Women received testosterone intramuscularly, and PBMCs were collected at different time points from the start of hormone treatment (0 (Days −9–0); 1 (Days 39–42); 2 (Days 112–126); and 3 (Days 201–219)) and stimulated with LPS (0.1 μg/mL) for 17 h. Concentrations of **a** TNF, **b** CCL2, and **c** CXCL1 in the supernatant were measured using a Multiplex Cytokine Assay or ELISA (CCL2). Each donor is depicted in an individual color (**a** $n_{0,1,2,3} = 5$; **b** $n_{0,1,2,3} = 5$; **c** $n_{0,1,2,3} = 5$). Correlation between testosterone and 5α-dihydrotestosterone (DHT) plasma levels in transgender men with **d** TNF, **e** CCL2, and **f** CXCL1 before transformation and at time point three post-transformation. Treatment of isolated peripheral, MACS-purified monocytes from healthy men and women with *E. histolytica* lysate (0.1 mg/mL) either alone or in the presence of DHT (10 nM) for 24 h. **g** TNF, **h** CCL2, and **i** CXCL1 concentrations in the supernatant were determined using a Multiplex Cytokine Assay or ELISA (CCL2) (**g** $n_{m/f\ DHT+/Ehis-} = 6/5$; $n_{m/f\ DHT-/Ehis+} = 6/5$; $n_{m/f\ DHT+/Ehis+} = 6/5$; **h** $n_{m/f\ DHT+/Ehis+} = 6/4$; $n_{m/f\ DHT-/Ehis+} = 6/4$; $n_{m/f\ DHT+/Ehis+} = 6/4$; **i** $n_{m/f\ DHT+/Ehis-} = 6/5$; $n_{m/f\ DHT-/Ehis+} = 6/5$; $n_{m/f\ DHT+/Ehis+} = 6/5$; pooled data from three independent experiments; Depicted are the means ± SEM; to normalize the baseline of the donors, values for unstimulated controls were subtracted from stimulated samples). **a–c** *P*-values were calculated using two-tailed paired analysis Student's *t* test (*$P < 0.05$) or **d–f** two-tailed Pearson correlation coefficient (*r*; *$P < 0.05$, **$P < 0.01$) with a simple linear regression. For **g–i** a two-tailed paired analysis was performed for each donor for each stimulation set up and two-tailed grouped analysis for the comparison between men and women: Student's *t*-test, *$P < 0.05$, **$P < 0.01$. Source data are provided as a Source Data file.

Here, we show that depleting CXCL1 reduces the liver damage during infection, a phenomenon paralleled by reduced frequency of Ly6C$^{hi}$ monocytes. Thus, both CXCL1 and TNF drive immunopathology during *E. histolytica* infection, especially in males.

It is important to identify parallels between murine and human data; therefore, we analyzed monocyte subpopulations from healthy men and women, and focused on the candidate sex-biased cytokines TNF, CXCL1, and CCL2. PBMCs were stimulated ex vivo with the TLR4-ligand LPS and an *E. histolytica* antigen preparation, which contains the TLR4-stimulant lipopo-peptidephosphoglycan[36]. Even though we found no sex differences after stimulation with the amebic lysate, classical monocytes from men showed significantly higher expression of CXCL1 after LPS stimulation. A recent transcriptomic approach by Schmiedel et al.,[53] who compared 54 male and 37 female samples, revealed that male-derived classical monocytes express more TNF than female-derived classical monocytes. Although classical monocytes

were the primary source of CXCL1, they observed no sex bias in CXCL1 expression under steady state conditions. Interestingly, female-derived classical monocytes expressed higher levels of CCL2[53], suggesting that sex-specific expression of cytokines may be modulated upon infection or by other challenges to the immune system. In agreement with the findings of Schmiedel et al. and ourselves (Table S1), earlier publications show that LPS-stimulated male classical monocytes express more TNF[9,54], even though we observed only a slight tendency toward increased protein expression by male-derived monocytes upon LPS stimulation. However, further HSNE analysis revealed for the first time that male-derived monocytes tend to express more cytokines simultaneously than female-derived monocytes, independent of the exogenous stimulant.

We questioned whether the potential to express TNF, CCL2, and CXCL1 simultaneously may have functional consequences due to autocrine binding of TNF to TNF receptors expressed by monocytes. Indeed, the amount of CXCL1, but not CCL2, in the

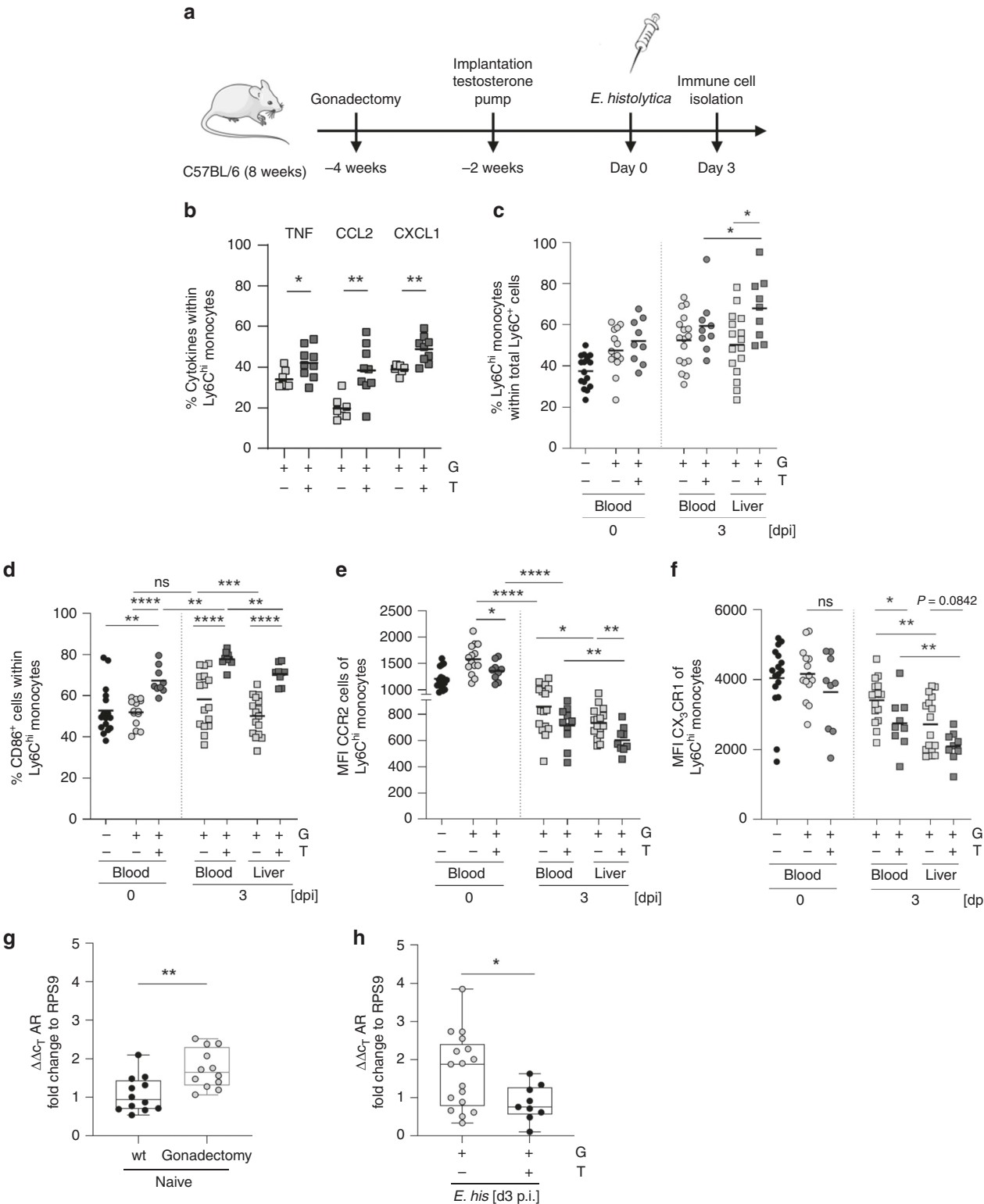

supernatant of LPS-stimulated isolated classical monocytes fell in the presence of TNF, TNFRI, and/or TNFRII-neutralizing antibodies. Previous studies suggest that TNF-induced expression of CXCL1 by endothelial and epithelial cell lines requires p38 MAPK, PI-3, Akt, JNK, and AP-1[27,55]. Using isolated classical monocytes, we report for the first time that p38 MAPK, PI-3K, Akt, and AP-1 are important checkpoints for induction of CXCL1 and CCL2 transcription and release by monocytes. Furthermore,

Cai et al.[40] depicted the influence of androgens on the induction of proinflammatory cytokine release by the sex hormone binding globulin (SHBG) and SHGB receptor complexes as well as non-voltage gated increase of $Ca^{2+}$ influx due to androgen-stimulated G-protein coupled receptor activation[40]. Both SHBG/SHBGR as well as the increased $Ca^{2+}$ influx have the potential to lead to enhance proinflammatory cytokine expression via the the recruitment of protein kinase A (PKA) and NFκB leading to a

**Fig. 7 Testosterone modulates classical monocytes towards a proinflammatory phenotype. a** Mice were either gonadectomized (G) or gonadectomized (4 weeks before infection, 8 weeks of age) and substituted with testosterone (T) (2 weeks before infection, 10 weeks of age). **b** Intracellular expression of TNF, CCL2, and CXCL1 by liver-derived Ly6C$^{hi}$ monocytes from G and T male mice 3 days following intrahepatic *E. histolytica* infection, as analyzed by flow cytometry. (TNF: $n_{G/T} = 7/9$; CCL2: $n_{G/T} = 7/9$, CXCL1: $n_{G/T} = 7/9$; one experiment; depicted are the means). **c** Percentage of Ly6C$^{hi}$ monocytes in the blood and liver of naïve or *E. histolytica*-infected G and T male mice (blood: $n_{wt/G/T\ d0} = 16/14/9$; $n_{G/T\ d3} = 16/9$; liver: $n_{G/T\ d3} = 16/9$; one experiment; depicted are the means). Percentage of **d** CD86$^+$ cells and **e** MFI of CCR2 and **f** CX$_3$CR1 Ly6C$^{hi}$ monocytes in the blood and liver of naïve or *E. histolytica*-infected G and T male mice (**d** blood: $n_{wt/G/T\ d0} = 16/14/9$; $n_{G/T\ d3} = 16/9$; liver: $n_{G/T\ d3} = 16/9$; **e** blood: $n_{wt/G/T\ d0} = 16/14/9$; $n_{G/T\ d3} = 16/9$; liver: $n_{G/T\ d3} = 16/9$; **f** blood: $n_{wt/G/T\ d0} = 16/14/9$; $n_{G/T\ d3} = 16/9$; liver: $n_{G/T\ d3} = 16/9$; one experiment; depicted are the means). **g** qPCR analysis of androgen receptor (AR) expression of RNA isolated from PBMCs of naïve sham operated or gonadectomized mice ($n_{sham/G} = 12$; one experiment; depicted are box ranging from 25th to 75th percentile and whiskers from min to max including the median). **h** qPCR analysis of androgen receptor (AR) expression of RNA isolated from PBMCs of *E. histolytica*-infected G and T mice ($n_{G/T} = 17/9$; pooled data from two independent experiments; depicted are box ranging from 25th to 75th percentile and whiskers from min to max including the median). *P*-values were calculated using two-tailed grouped analysis: Student's *t*-test (**b**) or Mann–Whitney test (**c, d, e–h**), *$P < 0.05$; **$P < 0.01$; ***$P < 0.001$; ****$P < 0.0001$. Source data are provided as a Source Data file.

potential further increase of CXCL1 by classical monocytes. However, in contrast to other cell types, we found that inhibiting JNK increased secretion of CXCL1, but not CCL2, by classical monocytes.

Ly6C$^{hi}$ monocytes are recruited via the CCR2/CCL2 axis. Here, we found that under steady-state conditions the percentage of Ly6C$^{hi}$ monocytes in the blood of male mice was higher than that in female mice; Kay et al.[56] reported similar observations for splenic cells. Despite a trend toward higher percentages of the corresponding classical monocyte subset in humans, we could not reproduce the data published by Jiang et al.[57]; likely due to different gating strategies (we included HLA-DR positive cells to exclude NK cells, as suggested by Heimbeck et al.)[19]. However, the higher frequency of circulating male-derived Ly6C$^{hi}$ monocytes observed in naïve mice and during *E. histolytica* infection was also observed in the liver during hepatic amebiasis. Regarding other typical monocyte surface markers for inflammation and recruitment, we found that naïve mice and classical monocytes from healthy human donors displayed no sex-specific differences in expression of CD86 and CX$_3$CR1, although male-derived classical monocytes showed higher expression of CCR2 in both species. Interestingly, classical murine bone-marrow-derived monocytes from males also express more CCR2 than those from females, suggesting that males already have a larger monocyte reservoir for egress; this may predispose males to higher numbers of classical monocytes in the circulation. With regard to the higher frequency of Ly6C$^{hi}$ monocytes at the site of infection in male mice, we postulate that further additional (although not yet precisely characterized) chemokine gradients may play a role in recruitment[23,24,58]. Human transcriptomic data analysis revealed that LPS-stimulated classical male-derived monocytes show markedly more chemoattractant GO term profiles, although additional receptors involved in recruitment were not identified explicitly.

During infection with *E. histolytica*, mice showed differential expression of CD86 and CX$_3$CR1. Expression of CD86 in both male- and female-derived classical monocytes in the blood increased upon infection. However, circulating and hepatic Ly6C$^{hi}$ monocytes from males showed higher expression of CD86, indicating a greater potential for activating subsequent adaptive immune responses[59]. In contrast to CD86, females express more CX$_3$CR1; however, its role remains unclear[32,60]. In this context, it is perhaps most likely that the CX$_3$CR1-CX$_3$CL1 interaction supports anti-inflammatory responses and tissue homeostasis in the liver, leading to a better disease outcome for females[16,61].

In the first set of experiments, we wanted to analyze the impact of androgens on expression of TNF, CCL2, and CXCL1 in humans. We were fortunate to have access to a transgender cohort of women transitioning to men under testosterone

treatment. After ~200 days of regular intramuscular testosterone injection, we analyzed the cytokines in the supernatant of LPS-stimulated PBMCs isolated from these individuals. The concentrations of TNF and CXCL1 increased over time. This was reflected by a strong correlation between these cytokines and plasma testosterone and DHT. Expression of CCL2 was unaffected and did not correlate with peripheral androgen levels, suggesting that CCL2 expression, in contrast to that of TNF and CXCL1, is not modulated by testosterone.

Detailed immunological studies in individuals undergoing gender transformation are rare. However, an earlier study reported that androgen treatment had little effect on immunological parameters. Interestingly, one study shows that TNF secretion by PHA-stimulated T cells from transgender men is not upregulated, whereas analysis of the supernatant of PHA-stimulated PBMCs revealed increased concentrations of TNF[62]. This suggests that other cells, including monocytes, rather than T cells are responsible for increased TNF concentrations. In line with this, androgens suppress expression of IFNγ by antigen-specific T cells and IL-12 signaling[7,63]. Human data show that androgen treatment increases CD4$^+$CD25$^+$Foxp3$^+$ regulatory T cell numbers[64], highlighting the regulatory function of androgens on adaptive immune responses.

To analyze the effects of androgens on isolated classical monocytes from men and women, we exposed them to the *E. histolytica* antigen preparation in the presence of the stronger (compared with testosterone) AR ligand DHT. Unlike testosterone, DHT is not converted to estradiol by aromatases; therefore, we excluded the possibility of monocyte modulation via estrogen receptor engagement. Interestingly, DHT increased production of CXCL1, but not the other cytokines, by male- and female-derived monocytes. Finally, to analyze the systemic effects of testosterone on monocyte populations, cytokines, and surface marker expression, we returned to the mouse model of amebiasis and substituted previously gonadectomized mice with testosterone prior to infection with *E. histolytica*. We found that we could mimic the initially observed differences between male and female mice, i.e., higher expression of TNF and CXCL1 by classical monocytes and higher frequencies of Ly6C$^{hi}$ monocytes in the liver upon testosterone treatment. As in male mice, testosterone-treated orchiectomized mice showed higher expression of CD86 and reduced expression of CX$_3$CR1 by classical monocytes in the blood on Day 3 post-*E. histolytica* infection. CD86 was higher expressed on hepatic classical monocytes from testosterone-treaded mice upon infection, whereas the expression of CX$_3$CR1 was only slightly reduced compared to the control group. However, expression of CCR2 appeared contrary to the original sex difference. Recent studies report a negative correlation between expression of CCR2 and intracellular AR expression in prostate cancer[43,44,65]. Here, we found that AR expression (at least by

PBMCs) in gonadectomized mice increased after castration, a putative counter-regulatory mechanism that compensates for decreased testosterone levels. By contrast, testosterone treatment reduced AR expression to levels seen in intact males.

In summary, we characterized the immune response underlying a classical, male-biased parasitic disease and identified a prominent role of classical monocytes expressing a typical cytokine set. Furthermore, there were similarities between peripheral murine and human classical monocytes with respect to the expression of surface markers and cytokine production during infection and after ex vivo stimulation, respectively. Finally, we found that CXCL1 is specifically upregulated by classical monocytes from mice and humans in an androgen-sensitive manner.

## Methods

**Human samples**. All human studies complied with all relevant ethical regulations. Experiments with blood samples from healthy Caucasian donors of both sexes (25–49 years of age) were approved by the ethics committee of the medical association Hamburg (Ethik-Kommission der Ärztekammer Hamburg; permission number: PV5252). All experiments were conducted under donor anonymization and in accordance with the relevant guidelines. The study including experiments with blood of females transitioning to males (transmales) was approved by the medical association Hamburg (Ethik-Kommission der Ärztekammer Hamburg; permission number: PV5245). The participants were recruited at the medical practice Amedes Experts in Hamburg, Germany and received regular intramuscular testosterone injections of 1000 mg. Written informed consent was obtained from all participants.

**Mice**. All murine studies complied with all relevant ethical regulations for animal testing and research. Animal experiments were performed under the agreement of the German animal protection law and were reviewed by the federal health authorities of the State of Hamburg (Behörde für Gesundheit und Verbraucherschutz; permission numbers 43/13 and 51/17) in accordance with the ARRIVE guidelines. C57BL/6J mice were either bred in the animal facility of the Bernhard Nocht Institute for Tropical Medicine or purchased at Charles River (Stock #000664; Sulzfeld, Germany) and kept in individually ventilated cages under specific pathogen-free conditions at the Bernhard Nocht Institute for Tropical Medicine with a day/night cycle of 12 h at a humidity of 50–60% and a temperature between 21 and 22 °C. Male and female mice were kept separately as well as gonadectomized testosterone-treated male mice to only gonadectomized mice as testosterone may lead to aggressive behavior in mixed groups. Mice were euthanized using 20% $CO_2$ per cage per minute followed by cervical dislocation.

**Hepatic amebiasis monitoring by magnetic resonance imaging**. At the age of 12–14 weeks male and female C57BL/6 mice were intrahepatically infected as previously described[31,66] with $2 \times 10^5$ in vitro-cultured E. histolytica trophozoites of the highly pathogenic clone B2, generated from the cell line B (HM-1:IMSS)[31].

Magnetic Resonance Imaging (MRI) was performed to monitor the ALA formation, using a small animal 7 Tesla MRI scanner (BrukerClinScan) with a T2-weighted turbo spin echo (T2wTSE). For the MRI screening, the animals were under a constant inhalation of an isoflurane/$O_2$ mixture with a flow rate of 500 L/min using a nose mask. The abscess volume was calculated with the OsiriX DICOM Viewer using the selected region of interest (ROI) in each slice of the transversal section of the abdomen[67]. Alternatively, the abscess size was determined using photo analysis with ImageJ software measuring the length, width, and height of the abscess on Day 3 p.i.

**Gonadectomy and testosterone treatment**. In total, 8-week-old male C57BL/6J mice were gonadectomized or sham operated. For testosterone treatment, an osmotic pump (micro-osmotic pump, Model 2004, ALZET) containing 5 mg/mL testosterone diluted in 45% w/w 2-Hydroxypropyl)-β-cyclodextrin (carrier) was implanted subcutaneously 2 weeks after gonadectomy. As a control, pumps only containing carrier solution were implanted in gonadectomized mice. Gonadectomy as well as pump implantation were performed under isoflurane narcosis. Two weeks after testosterone treatment mice were infected with E. histolytica trophozoites intrahepatically.

**Isolation of murine immune cells**. Murine blood was directly taken with a heart puncture after euthanization of the mice and collected in EDTA-coated tubes. Immune cells were obtained after performing two erythrocyte lysis steps. Hepatic immune cells were isolated using a Percoll gradient. The homogenized liver was resuspended in 80% Percoll (GE Healthcare) and overlayed with 40% Percoll diluted in complete 1640 RPMI (Gibco) containing 10% fetal calf serum, 1% L-Glutamine, and 1% penicillin/streptavidin (cRPMI). After a centrifugation step without break (800×g; 25 min; 21 °C) the cells localized at the interphase of the two Percoll layers and were transferred into a new collection tube for further washing

steps with PBS and an additional erythrocyte lysis. The obtained single-cell suspension was washed twice with cRPMI.

**Isolation of human peripheral blood mononuclear cells**. Peripheral blood mononuclear cells (PBMCs) isolation from healthy human donors was performed using a Biocoll (Merck) gradient. The Biocoll solution was overlayed with 1:2 diluted blood in PBS. The centrifugation was carried out without break (450 × g; 30 min; 21 °C) and the cells localized at the interphase of the layers were isolated and washed for further procedures.

**Antibodies and flow cytometry**. CXCL1 neutralization in vivo was performed using 85 µg αCXCL1 (#MAB453, R&D) per mouse injected intra-peritoneally 1 day prior to and 1 day post E. histolytica-infection. Control mice were injected with the corresponding amount of rat IgG2A (#MAB006, R&D). FACS analyses were performed on single-cell suspensions of liver or blood-derived immune cells. For the live/dead discrimination using flow cytometry analysis, cells were stained according to manufacturer's instructions with either Zombie UV or Aqua (1:1000, #423108, #423102, BioLegend). For experiments involving intracellular staining of single-cell suspensions, $1 \times 10^6$ immune cells were stimulated in cRPMI containing 50 ng/mL PMA (Sigma-Aldrich) and 500 ng/mL ionomycin (Sigma-Aldrich) for 4 h including 5 µg/mL Brefeldin A (Sigma-Aldrich) for the last 3 h. Intracellular staining was performed using the Foxp3 transcription factor staining buffer (eBioscience). The following antibodies were used for murine lymphocyte staining and purchased at BioLegend, if not indicated otherwise: CD11b BV510 (1:400, M1/70), Ly6C FITC (1:300, HK1-4), Ly6C PerCP/Cy5.5 (1:300, HK1-4) Ly6C PE (1:900, HK1-4), Ly6G AF700 (1:400, 1A8), Ly6G APC/Cy7 (1:500, 1A8), Ly6G PE (1:800, 1A8) CD86 AF700 (1:100, GL-1), CCR2 PE/Cy7 (1:100, SA203G11), CX₃CR1 PerCP/Cy5.5 (1:200, SA011F11), TNF BV421 (1:200, MP6-XT22), CCL2 PE (1:50, 2H5), and CXCL1 Alexa Fluor 647 (1:50, 1174A, R&D).

For the flow cytometry analysis of human PBMCs, frozen PBMCs were thawed and left over night at 37 °C 5% $CO_2$ at a concentration of $1 \times 10^6$/mL in cRPMI. In all, $2 \times 10^6$ were stimulated for 6 h with either 0.1 µg/mL LPS or 0.1 mg/mL E. histolytica lysate including the last 5 h with 5 µg/mL Brefeldin A (Sigma-Aldrich). The staining of TNFRI, TNFRII, and the androgen receptor was only performed on freshly isolated PBMCs. The following anti-human antibodies were utilized for the characterization and were purchased at BioLegend, if not indicated otherwise: HLA-DR BUV 395 (1:20, G46-6, Becton Dickinson), CD14 Alexa Fluor 700 (1:20, M5E2), CD14 PerCP/Cy5.5 (1:20, M5E2), CD16 PerCP (1:20, 3G8), CCR2 Allophycocyanin (1:10, K036C2), CCR5 Alexa Fluor 488 (1:10, J418F1), CX₃CR1, PE/Cy7 (1:20, 2A9-1), TNFRI unconjugated (1:40, MAB225, R&D), TNFRII unconjugated (1:30, MAB726, R&D), and AR Alexa Fluor 488 (1:50, ER179(2), abcam). The appropriate isotype control, recombinant Rabbit IgG Alexa Fluor 488 (1:50, EPR25A, abcam) was performed for the verification of the AR Alexa Fluor 488 staining. The unconjugated TNFRI and TNFRII stainings including the secondary antibody incubation with anti-mouse IgG (H + L) Alexa Fluor 594 (1:400, #A-11005, ThermoFisher Scientific), were also verified using the appropriate isotype control (11711, MAB002, R&D).

Flow cytometry analysis was performed on a BD LSRFortessa™ (Becton Dickinson) and the data were analyzed with the FlowJo V10.4.2 software. Cluster analysis was performed using the HSNE algorithm by Cytosplore version 2.2.1[38,39]. The analysis was performed on CD14+ concatenated cells from five donors per sex and stimulation. The HSNE analysis was calculated in parallel for 1 male and 1 female sample including $1 \times 10^5$ events per sex, including the markers: CD64, CCR5, CCR2, CX₃CR1; CD16; TNF; CXCL1; and CCL2.

**Preparation of E. histolytica lysate**. E. histolytica trophozoites were harvested and centrifuged (250×g; 5 min; 4 °C). The supernatant was discarded and the pellet was frozen and thawed three times using liquid nitrogen. The protein concentration was measured with the Nanodrop™ 2000 spectrometer (Thermo Scientific).

**LPS-stimulation of PBMCs derived from transgender men**. PBMCs of women undergoing testosterone therapy (transgender men) were stimulated with 0.1 µg/mL LPS for 17 h. The supernatant was collected and stored at −80 °C for further cytokine analysis as described below.

**Androgen influence on monocyte stimulation**. Monocytes from men and women were isolated with the Pan monocyte Isolation Kit (#130-096-537, Miltenyi Biotec) and stimulated with either 0.1 mg/mL E. histolytica lysate 10 nM 5α-dihydrotestosterone (DHT) (Sigma-Aldrich) for 24 h. The supernatant was stored at −20 °C for further cytokine analysis as described below.

**Monocyte signaling pathway characterization for CXCL1**. The anti-Human CD14 Magnetic Particles (Becton Dickinson) were utilized for the purification of CD14+ monocytes. In all, $1 \times 10^5$ CD14+ monocytes were incubated with 25 ng/mL αTNF mAb (MAB210, clone 1825, R&D), 1 µg/mL αTNFRI (MAB225, clone 16803, R&D), 10 µg/mL αTNFRII (MAB726, clone 22110, R&D) or with the appropriate isotype control (MAB002, R&D) for 1 h at 37 °C, 5% $CO_2$, prior stimulation with 0.1 µg/mL LPS for additional 24 h and CXCL1 was determined in

the supernatant ELISA (described below). The impact of the MAPK pathway was analyzed as follows: $1 \times 10^5$ CD14$^+$ monocytes were incubated for 30 min in X-vivo$^{TM}$ serum-free medium (Lonza) with the following inhibitor compounds: 10 μM SP600125 (JNK, Sigma-Aldrich), 10 μM SB202190 (p38, Sigma-Aldrich), 10 μM LY294002 (PI3K, Sigma-Aldrich), 17 μM Tanshinone (AP-1, Sigma-Aldrich), 5 μM BAY11-7085 (NFκB, Santa Cruz Biotechnology) or 4 μM Actino-mycin D (DNA transcription, Sigma-Aldrich). After this pre-incubation the cells were stimulated for 16 h with 0.1 μg/mL LPS and CXCL1 was determined in the supernatant as described below.

**Cytokine and hormone analysis**. For cytokine analysis supernatant of stimulated cells or the blood of *E. histolytica*-infected mice was either taken by sub-mandibular blood collection or by heart puncture after euthanasia and collected in EDTA-coated tubes. The plasma was obtained after centrifugation (1000×*g*; 10 min; 4 °C) and stored at −20 °C for further cytokine analysis.

The cytokine analysis was performed using either murine or human customized LEGENDplex kits (BioLegend). Furthermore, concentration of CXCL1 or CCL2 in the supernatant of stimulated human cells was determined by using CXCL1/GRO alpha DuoSet ELISA development kit (R&D systems) or the MCP-1/CCL2 Deluxe Set (BioLegend), respectively.

Hormone concentration were analyzed out of 1 mL plasma of each transgender donor at given time points. Until analysis, plama samples were kept at −80 °C. The analysis was carried out by mass spectrometry using the MassChrom® Steroide im Serum/Plasma mit Sample Clean Up Columns - LC-MS/MS (ChromSystems).

**RT-PCR**. The RNA of frozen liver tissue or PBMCs was isolated using TRIzol reagent (Life Technologies) and subsequently reverse transcribed into cDNA using the Maxima® First Strand cDNA Kit (Thermo Scientific). The cytokine levesl were calculated using the $2^{-\Delta\Delta Ct}$ method with the ribosomal protein S9 (*RPS9*) as housekeeping gene. The following primers were utilized: fwd-AR: 5′-TGAGTAC CGCATGCACAAGT-3′, rev-AR: 5′-GCCCATCCACTGGAATAATGC-3′; fwd-CCL2: 5′-TCTCTCTTCCTCCACCACCA-3′, rev-CCL2: 5′-CGTTAACTGCATC TGGCTGA-3′; fwd-CXCL1: 5′-ACTCAAGAATGGTCGCGAGG-3′; rev-CXCL1: 5′-ACTTGGGGACACCTTTTAGCA-3′, fwd-RPS9:: 5′-CCGCCTTGTCTCTC TTTGTC-3′; and rev-RPS9:: 5′- CCGGAGTCCATACTCTCCAA-3′. The anneal-ing temperature for all genes of interest was set at 60 °C while running the RT-PCR with the Maxima SYBR Green qPCR Master Mix (Thermo Scientific) at the Roche Lightcylcler®.

**RNA sequencing and data analysis**. PBMCs were stimulated with 0.1 μg/mL LPS for 4 h at 37 °C and 5% CO$_2$. CD14$^+$ monocytes were isolated with the anti-Human CD14 Magnetic Particles Kit (Becton Dickinson) according to the instructions of the manufacturer. The purity of the cell isolation was verified by flow cytometry analysis using αCD14$^+$ PerCP/Cy5.5 (clone M5E2, BioLegend). Samples with a purity under 90% were excluded for further analysis. The isolated CD14$^+$ monocytes were resuspended in 500 μL TRIzol, incubated for 5 min at 37 °C and stored at −80 °C. The RNA isolation was carried out after thawing the samples on ice and mixing them with chloroform in a 1:5 ratio. After an incubation for 3 min at room temperature the samples were centrifuged at 12,000×*g* at 4 °C for 15 min and the supernatant was diluted with 70% ethanol to the same amount of volume. The RNA was further purified with the MinELute RNeasy Kit (Qiagen), including a DNAse (Qiagen) digestion step. The transcriptomic analysis was conducted only on RNA samples meeting the following criteria, verified at the Bioanalyzer (Agilent): RNA amount >200 ng; concentration >20 ng/μL; RIN > 7; 28S/18S > 1. The library preparation as well as the sequencing was performed by BGI Genomics, China. The data were paired end short reads, which were subse-quently matched to the human reference genome GRCH38.82. The aligning was performed with the software HISAT2 and gene counting was performed with the "featureCounts" module of the "subreads" software package. Differential tran-scriptome analyses were performed with the Deseq2 software.

The gene set analysis was performed using GOrilla[68,69] as well as PANTHER GO-slim[34] on genes meeting the following criteria: log2FoldChange > 2 p adjust (adj) < 0.05. The heatmap was created with the open source software "heatmapper"[70].

**Software**. The MRI screening was controlled by NUMARIS/4 syngo MR B15. Osirix Imaging Software Dicom Viewer Version 32-bit 5.6 and ImageJ 1.51n were used to analyze MRI-based imaging. For the LSRFortessa instrument control the FACSDiva V8 software was applied. Further flow cytometry data analysis was conducted with FlowJo V10.4.2 and Cytosplore version 2.2.1. LEGENDPlex V8.0 was applied to analyze performed cytometric bead assays. ELISA-readouts were performed using the Revelation software G 3.2. Quality check of isolated RNA was carried out with the Bioanalyzer Agilent Software. Analysis of RT-PCRs were conducted using the Roche Lightcycler® Software 1.1.1320. RNAseq data was conducted using HISAT2 and featureCounts including the subreads package. Deseq2 software was utilized to perform differential transcriptome analysis. Resulting data were further analyzed using open source softwares: heatmapper, Gorilla and Panther Version 15.0. To generate graphs and statistical analysis GraphPad Prism 8.4.2 was utilized and figures were configured with Gimp 2.10.20.

Cartoons and illustrations were created with PowerPoint 2013 and a licensed version of biorender.com.

**Statistical analysis**. All data were analyzed with the Shapiro–Wilk test for normal distribution prior testing for statistical differences between groups. Statistical analysis are indicated in each figure legend and were carried out using either a parametric paired or unpaired two-tailed Student's *t* test for normal distribution, or a nonparametric two-tailed Mann–Whitney test or Wil-coxon matched-paired signed rank test for non-normal distribution (GraphPad Prism V8.4.0). *P*-values are indicated as *$p \leq 0.05$, **$p \leq 0.01$, ***$p \leq 0.001$, ****$p \leq 0.0001$.

**Reporting summary**. Further information on research design is available in the Nature Research Reporting Summary linked to this article.

## Data availability

The data of this manuscript are available from the corresponding author. All other data are included in the supplemental information or available from the authors upon reasonable requests. The transcriptomic data in Fig. 3 have been deposited in NCBI SRA Archive under the accession_codes: SRX8527978 SRX8527979 SRX8527980 SRX8527981 SRX8527982 SRX8527983 SRX8527984 SRX8527985 SRX8527986 SRX8527987 SRX8527988 SRX8527989 SRX8527990 SRX8527991 and the bioprojectID PRJNA638783. The reference genome used for this study was GCHR32.82 [ftp://ftp.ensembl.org/pub/release-82/fasta/homo_sapiens/]. Source data are provided with this paper.

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

## Acknowledgements

We thank Malte Vogelsang and Nadine Kottmayr as well as the members of the animal facility for their support. Additionally we thank Madeleine Hamley for her helpful comments and edits of the manuscript. This study was enabled by grants of the State Research Funding - FV45, Authority for Science, Research and Equality, Hanseatic City of Hamburg, Germany.

## Author contributions

H.L. conceived the study; J.S., M.G., and H.L. designed and performed experiments and J.S. and H.L. wrote the manuscript; H.F., S.H., C.M., C.H., S.K., J.N., M.W., V.W., S.H.H., C.D., N.G.M. S.S.B., and J.W. performed experiments and data analysis; T.T. analyzed transcriptomic data; H.I., E.T., I.B., T.J., and M.A. edited the manuscript.

## Competing interests

The authors declare no competing interests.

## Additional information

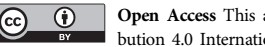

