## [Peer Review File · Nature Communications]

Reviewers' comments NCOMMS-19-26999

Reviewer #1 (Amebic liver abscess, amebiasis) (Remarks to the Author):

The authors address one of the biggest mysteries in parasitology, which is why amebic liver abscess is predominantly (80-90%) a disease of young adult males. A male predominance of infection is also seen in several other infections and diseases, making the work presented here of broader significance. Prior work in the murine model of amebic liver abscess demonstrated that liver damage in hepatic amebiasis is due to CCL2 recruited monocytes that produce TNF alpha.

Here the authors demonstrate that classical monocytes from males express higher levels of CCR2 and secrete higher levels of CXCL1. In contrast monocytes from females stimulated with androgens led to increased expression of CXCL1. Neutralization of CXCL1 in the mouse model decreased abscess size in the liver demonstrating a causal role of CXCL1.

1) In Figure 1 is there a reduction in Ly6c hi monocytes in the liver abscess from a CXCL1-neutralized mouse?

2) Is the reduction in CCR2 expression in monocytes seen in females of functional significance (chemotaxis in a gradient of CCL2)?

Reviewer #2 (Hormone/immune crosstalk) (Remarks to the Author):

In this article, Sellau et. al. used human PBMCs primarily stimulated with the TLR4 ligand, LPS, and a mouse model of hepatic amebiasis to understand the role of monocytes in driving the differences in immune responses and immunopathology between males and females. The authors show that in humans as well as in a mouse model of parasitic infection, male-derived classical monocytes have greater chemoattractant properties and secrete more pro-inflammatory cytokines than female-derived cells, which likely impacts sex differences in disease pathogenesis. The authors go on to show that androgens play an important role in drive this response of monocytes following stimulation. My specific comments are outlined below.

Major:

1. There is poor connection between the experiments. There is no doubt that the authors have done a lot of work. Missing, however, is the logical progression of the studies, which combine mouse parasitic infections to examine sex differences, human PBMC studies to evaluate sex differences, human cells in vitro and transgender in vivo studies to consider androgens, and the use of a parasite that likely does not exclusively signal through TLR4 in mouse studies combined with the use of LPS, which does signal through TLR4, in human studies. There are a lot of data that suggest interactions, but no unification across studies. This might be improved by reorganizing the results. For example, the human in vitro data could be presented first, with the transgender male study presented in separate section, and mouse study placed at the end to show functional significance of these cells and effects of androgens (see below) in a disease pathogenesis model. The abstract reflects this problem and reads as a list of unrelated findings from a lot of different studies.

2. The direct evidence suggesting the role of androgens in hepatic amebiasis is missing and results in this study not fitting the theme of this paper (even the title of the paper). Hormonal manipulations such as inclusion of gonadectomized mice (absence/reduction of testosterone) treated with testosterone or DHT would further help to strengthen the argument that androgens predispose males to enhanced immunopathology by providing.

3. Differential use of testosterone versus DHT in human studies. In the studies using blood from transgender males, these individuals were taking testosterone (Fig 4d-f), which can signal through both the estrogen and androgen receptor. In contrast, the studies of PBMCs from 'health' donors involved isolating and treating cells in vitro with DHT (Fig 4g-i), which signals exclusively through the androgen receptor. The authors need to resolve this experimental inconsistency.

Minor:

1. Detailed information is lacking in multiple places. Texts and the figure legends should explain in detail about the sex of the animal used (e.g. 1J, K) and the age of men and women enrolled in the study.

Reviewer #3 (Innate immune cell trafficking, chemotaxis)(Remarks to the Author):

Authors have examined role of androgens in monocyte responses to LPS and amebic antigens. Some of the results are interesting but the results and conclusions are problematic.

1. Many of the figures contain only 2 (two) independent experiments and still parametric statistics are used (Student t test). Also the differences are very small in many figures (fig 1h, 2c, 3b, 3b, 3e, 5b) that do not warrant the conclusions stated by the authors especially considering the few experiments.

2. Data on expression of molecules related to leukocyte migration do not warrant conclusion about actual migration. Effects on migration must be studied directly in order to make conclusions about it.

3. It is unclear why the authors used LPS when wanting to make conclusions about amebic infection. Also the LPS protocols vary throughout the manuscript without explanation.

4. There are a lot of conclusions and speculations that should be deleted from the result section.

Revision 1 NCOMMS-19-26999

Reviewers' comments :

Reviewer #1 (Amebic liver abscess, amebiasis) (Remarks to the Author):

The authors address one of the biggest mysteries in parasitology, which is why amebic liver abscess is predominantly (80-90%) a disease of young adult males. A male predominance of infection is also seen in several other infections and diseases, making the work presented here of broader significance. Prior work in the murine model of amebic liver abscess demonstrated that liver damage in hepatic amebiasis is due to CCL2 recruited monocytes that produce TNF alpha.

Here the authors demonstrate that classical monocytes from males express higher levels of CCR2 and secrete higher levels of CXCL1. In contrast monocytes from females stimulated with androgens led to increased expression of CXCL1. Neutralization of CXCL1 in the mouse model decreased abscess size in the liver demonstrating a causal role of CXCL1.

1. In Figure 1 is there a reduction in Ly6c^{hi} monocytes in the liver abscess from a CXCL1-neutralized mouse?

Yes, that is correct. CXCL1-depleted mice have a reduced Ly6C^{hi} monocyte subpopulation (please see Fig. 1I).

2. Is the reduction in CCR2 expression in monocytes seen in females of functional significance (chemotaxis in a gradient of CCL2)?

This is a very good question; however, it is widely accepted that the CCR2-CCL2 interaction is responsible for egress of monocytes from the bone marrow to the circulation. Hence, we find already in naive animals a sex bias toward males; this is true in both the circulation (Fig. 2c) and the bone marrow (Supplementary 1c). This indicates preferential recruitment via this CCR2-CCL2 axis from the bone marrow^{1,2}. However, it is controversial whether a CCL2 gradient also leads to recruitment of monocytes to sites of inflammation or injury since several chemokine recruitment factors play a role in this¹⁻⁴.

Reviewer #2 (Hormone/immune crosstalk) (Remarks to the Author):

In this article, Sellau et. al. used human PBMCs primarily stimulated with the TLR4 ligand, LPS, and a mouse model of hepatic amebiasis to understand the role of monocytes in driving the differences in immune responses and immunopathology between males and females. The authors show that in humans as well as in a mouse model of parasitic infection, male-derived classical monocytes have greater chemoattractant properties and secrete more pro-inflammatory cytokines than female-derived cells, which likely impacts sex differences in disease pathogenesis. The authors go on to show that androgens play an important role in drive this response of monocytes following stimulation. My specific comments are outlined below

Major:

1. There is poor connection between the experiments. There is no doubt that the authors have done a lot of work. Missing, however, is the logical progression of the studies, which combine mouse parasitic infections to examine sex differences, human PBMC studies to evaluate sex differences, human cells in vitro and transgender in vivo studies to consider androgens, and the use of a parasite that likely does not exclusively signal through TLR4 in mouse studies combined with the use of LPS, which does signal through TLR4, in human studies. There are a lot of data that suggest interactions, but no unification across studies. This might be improved by reorganizing the results. For example, the human in vitro data could be presented first, with the transgender male study presented in separate section, and mouse study placed at the end to show functional significance of these cells and effects of androgens (see below) in a disease pathogenesis model. The abstract reflects this problem and reads as a list of unrelated findings from a lot of different studies.

Thank you for the comments concerning the reorganization of the manuscript, which we found to be very helpful. However, our longstanding work and expertise in the area of hepatic amebiasis led us to focus on the dominant role of classical monocytes, the CCL2-CCR2 axis, and TNF α in the immunopathology underlying hepatic amebiasis in the murine model ⁵. This provided the basis for us to further examine the contribution of monocytes to sex-based differences in disease. Nevertheless, we have reorganized the result section to clarify the connection between the murine and human data sets.

We have separated the now more detailed characterization of murine classical monocytes from the first figure (which is now Figure 2). We then move on to characterization of human monocyte subpopulations. Beginning with the distribution pattern of monocyte subpopulations between men and women, we analyzed the transcriptome profile of classical monocytes from men and women; this led us to characterize the factors responsible for recruitment of these cells.

Due to striking similarities in sex-specific cytokine expression between murine and human classical monocytes, we next investigated the influence of testosterone on cytokine production in individual humans and specifically in monocytes from individual humans (see detailed discussion of your remark #3). To provide new empirical evidence for androgens raised from our human data, we now present a new data set regarding hormonal manipulation in the mouse model for hepatic amebiasis, with specific focus on classical monocytes. Concerning TLR4 signaling, we are aware that *E. histolytica* does not signal exclusively via TLR4; however, our antigen preparation contains a TLR4 activator ⁶, lipopeptidophosphoglycan, a molecule that we are focusing on in our lab ⁷. We hope that this reorganization helps to improve the logical structure of the manuscript.

2. The direct evidence suggesting the role of androgens in hepatic amebiasis is missing and results in this study not fitting the theme of this paper (even the title of the paper). Hormonal manipulations such as inclusion of gonadectomized mice (absence/reduction of testosterone) treated with testosterone or DHT would further help to strengthen the argument that androgens predispose males to enhanced immunopathology by providing.

Thank you for this suggestion. As requested, we have performed additional animal experiments. We examined the effect of gonadectomy and testosterone substitution on

expression of cytokines, surface markers, and localization of classical monocytes in the murine model of hepatic amebiasis (see Figure 7a–f).

3. Differential use of testosterone versus DHT in human studies. In the studies using blood from transgender males, these individuals were taking testosterone (Fig 4d-f), which can signal through both the estrogen and androgen receptor. In contrast, the studies of PBMCs from 'health' donors involved isolating and treating cells *in vitro* with DHT (Fig 4g-i), which signals exclusively through the androgen receptor. The authors need to resolve this experimental inconsistency.

You are correct; aromatase can convert testosterone to estradiol, leading to signaling via the estradiol receptor. Therefore, we analyze the levels of both testosterone and DHT in the serum of the transgender cohort. We found a significant positive correlation between testosterone/DHT and TNF α and CXCL1, although DHT was more pronounced (Figure 6d–f).

For the same reason, we used DHT in the *in vitro* studies since it is: a) the most active androgen, and b) not converted to other hormones. We have mentioned this in the Result section (see page 8, lines 259-261) and Discussion section (see page 13, lines 420-422).

Minor:

1. Detailed information is lacking in multiple places. Texts and the figure legends should explain in detail about the sex of the animal used (e.g. 1J, K) and the age of men and women enrolled in the study.

The reviewer is correct. We have provided the relevant information in the figure legends.

Reviewer #3 (Innate immune cell trafficking, chemotaxis)(Remarks to the Author):

Authors have examined role of androgens in monocyte responses to LPS and amebic antigens. Some of the results are interesting but the results and conclusions are problematic.

1. Many of the figures contain only 2 (two) independent experiments and still parametric statistics are used (Student t test). Also the differences are very small in many figures (fig 1h, 2c, 3b, 3b, 3e, 5b) that do not warrant the conclusions stated by the authors especially considering the few experiments.

We agree. The figure legends have been revised accordingly. Moreover, we repeated many of the experiments using a larger number of samples (see Fig. 1b, h–l; Fig. 2a–d; Fig. 3b–e; and Fig. 4b–d). The number of independent experiments was also increased. In addition, we have included Figure 7, which is based on two independent experiments with $n = 8–14$ animals.

However, samples (especially those from transgender individuals (TG individuals) or from animal experiments) are valuable and rare, so we kept sample numbers per experiment as low as possible while still complying with the ARRIVE guidelines.

We agree that the Student's t test requires a normal sample distribution, which is difficult to achieve when small sample numbers are available. Therefore, we used the Shapiro-Wilk test, which allows us to test whether the underlying numbers of a sample group is

normally distributed. This test is characterized by its comparatively high test strength, especially when testing smaller samples ($n < 50$). Hence, we re-analyzed all data from our experiments using the Shapiro-Wilk test (GraphPad Prism statistic software version 8.4.0) before applying Student's t test. If the distribution was not normal, we used the Mann-Whitney U test. The tests applied are stated in the respective Figures and the Materials and Methods section (page 18, lines 613-618).

2. Data on expression of molecules related to leukocyte migration do not warrant conclusion about actual migration. Effects on migration must be studied directly in order to make conclusions about it.

This comment is of special importance; however, we found sex differences in expression of CCR2 by monocytes in the bone marrow, which explains the higher levels in the circulation of male individuals (see. Suppl. Fig). CCL2-dependent recruitment is primarily responsible for the egress of CCR2⁺ monocytes from the bone marrow; however, it is unclear whether a CCL2 gradient recruits classical monocytes to sites of inflammation or injury.¹⁻⁴

Therefore, throughout the manuscript we have toned down statements concerning migration and recruitment of monocytes.

3. It is unclear why the authors used LPS when wanting to make conclusions about amebic infection. Also the LPS protocols vary throughout the manuscript without explanation.

Both LPS and *E. histolytica* antigens, especially lipopeptidophosphoglycan, signal via TLR4⁶.

We think that this justifies the use of the two different stimuli. First, higher CXCL1 expression by male-derived classical monocytes is independent of species (Fig. 1 j; Fig. 4d); second, and as shown by HSNE analysis, we found a near-congruent distribution of cytokine-producing classical monocytes in men and women, independent of stimulation with LPS or *E. histolytica* antigen preparation (Fig. 4e,f).

In all experiments, the concentration of LPS was 0.1 $\mu\text{g/mL}$.

For FACS analysis, we stimulated cells for only 6 hours to maintain integrity of surface marker expression pattern.

LPS stimulation of PBMCs from the transgender cohort was performed in the lab of M. Altfeld, while *ex vivo* stimulation of monocytes with LPS (Fig.5) was performed in our own lab.

We have provided more detailed information in the respective figure legends (Fig. 5a–f; Fig. 6a–c and g–i) and revised the text in the Material and Methods (page 16 line 523, page 16 lines 547-549, page 17 lines 560-561).

4. There are a lot of conclusions and speculations that should be deleted from the result section.

As you suggest, we have tried to remove all speculative statements from the Results section; also, we have tried to keep all statements as neutral and concrete as possible.

Revision 2 NCOMMS-19-26999A

REVIEWERS' COMMENTS:

Reviewer #1 (Remarks to the Author):

The authors have comprehensively answered the concerns from the prior review and the manuscript is improved as a result. I have no further critiques.

Reviewer #2 (Remarks to the Author):

The authors adequately addressed my major concerns.

A few modifications that the authors should consider:

1. I would not use a slang term like 'trans' in a manuscript. Replace with 'transgender'.
2. In Figure 5g, the conceptual cartoon image would benefit from showing how androgen receptor signaling interacts with TLR4 signaling to make the cartoon more relevant to this manuscript.
3. In Figure 7d and related text, I would suggest replacing 'lymphocyte isolation' with 'immune cell isolation' because monocytes are not lymphocytes.

Reviewer #3 (Remarks to the Author):

The manuscript has improved. However, to still use student t test for small sample groups is not acceptable and inconsistent even if the authors now used another test to evaluate potential normal distribution in small samples. Non-parametric test must be used for these groups with less than 10-20 animals/independent samples.

Reviewers' comments :

Reviewer #1 (Remarks to the Author):

The authors have comprehensively answered the concerns from the prior review and the manuscript is improved as a result. I have no further critiques.

We thank the referee for the supportive comment.

Reviewer #2 (Remarks to the Author):

The authors adequately addressed my major concerns.

A few modifications that the authors should consider:

1. I would not use a slang term like 'trans' in a manuscript. Replace with 'transgender'.

This is a good remark. We changed the term 'trans' throughout the manuscript.

2. In Figure 5g, the conceptual cartoon image would benefit from showing how androgen receptor signaling interacts with TLR4 signaling to make the cartoon more relevant to this manuscript.

This point clearly leads to an improved version of the manuscript and we created a new cartoon including the putative impact of androgens on the TLR4 signaling pathway leading to the transcription of the cytokines CCL2 and CXCL1.

Fig. 5g: Proposed scheme outlining differences in CCL2 and CXCL1 expression by LPS-stimulated CD14⁺ monocytes, Based on the study by Lo et al. (2014)⁸, Cai et al. (2016)⁹ and Koryankina et al. (2014)¹⁰ (green line=part of the signal transduction for CCL2 and CXCL1, blue interrupted line=inhibition of CXCL1 expression, yellow line=induces CCL2, black line=induced by androgens, grey dotted line=interaction assumed).

3. In Figure 7d and related text, I would suggest replacing 'lymphocyte isolation' with 'immune cell isolation' because monocytes are not lymphocytes.

Likewise to the first point, we are thankful for this remark and changed the the phrase not only in Fig. 7, but also in the rest of the manuscript.

Reviewer #3 (Remarks to the Author):

The manuscript has improved. However, to still use student t test for small sample groups is not acceptable and inconsistent even if the authors now used another test to evaluate potential normal distribution in small samples. Non-parametric test must be used for these groups with less than 10-20 animals/independent samples.

We are aware that small sample sizes are hard to test for normal distribution and hence for significance testing in general, but we trust in the very well validated and numerous published statistical methods for the analysis of normality.

In our study we analyzed all data using the Shapiro-Wilk test, which is known as one of the most powerful tests for normality, allowing to perform normality testing starting with only three samples¹¹. For sample sizes under n=50 it is recommended in general to use the Shapiro-Wilk test before analyzing for significant differences between two groups. If the test does not reject normality, this suggests that a parametric procedure that assumes normality like e.g. a t-test can safely be used¹².

Parametric statistical tests only work for normal, continuously distributed data, whereas non-parametric methods are free of this assumption and work therefor well for data with skewed distributions. The purpose of the t-test is to compare certain characteristics representing groups, and the mean values become representative when the population has a normal distribution. The hypotheses used in testing our data normality are as follows:

H₀: The distribution of the data is normal.

H₁: The distribution of the data is not normal.

The normality assumption means that the collected data follows a normal distribution, which is essential for parametric assumption. If the null hypothesis of normality is rejected, a non-parametric test should be applied. Whether the null hypothesis is rejected or true can be answered looking at the P values from the normality test. In our analysis a P value greater than 0.05 answers the null hypothesis with Yes. If the P value is less than or equal to 0.05, the null hypothesis is rejected and the alternative hypothesis is accepted. In our study this was decisive to perform the non-parametric Mann-Whitney U test. This can be controlled looking at our raw data sets (Sellau et al. 2020_source data file). Additionally, we tried to point this out in our material and methods section.

We would like to mention that using the t-test is widely accepted if tested for normal distribution with Shapiro-Wilk, and is applicable even with small sample sizes^{13,14}. We hope that we could give a better understanding of why we have compiled our statistics in this way.

References

1. Serbina NV, Pamer EG. Monocyte emigration from bone marrow during bacterial infection requires signals mediated by chemokine receptor CCR2. *Nature immunology* **7**, 311-317 (2006).
2. Shi C, Pamer EG. Monocyte recruitment during infection and inflammation. *Nature reviews Immunology* **11**, 762-774 (2011).
3. Winkler CW, et al. Cutting Edge: CCR2 Is Not Required for Ly6C(hi) Monocyte Egress from the Bone Marrow but Is Necessary for Migration within the Brain in La Crosse Virus Encephalitis. *Journal of immunology (Baltimore, Md : 1950)* **200**, 471-476 (2018).
4. Gschwandtner M, Derler R, Midwood KS. More Than Just Attractive: How CCL2 Influences Myeloid Cell Behavior Beyond Chemotaxis. *Frontiers in Immunology* **10**, (2019).
5. Helk E, et al. TNFalpha-mediated liver destruction by Kupffer cells and Ly6Chi monocytes during *Entamoeba histolytica* infection. *PLoS pathogens* **9**, e1003096 (2013).
6. Wong-Baeza I, et al. The role of lipopeptidophosphoglycan in the immune response to *Entamoeba histolytica*. *J Biomed Biotechnol* **2010**, 254521-254521 (2010).
7. Lotter H, et al. Natural killer T cells activated by a lipopeptidophosphoglycan from *Entamoeba histolytica* are critically important to control amebic liver abscess. *PLoS pathogens* **5**, e1000434 (2009).
8. Lo HM, Lai TH, Li CH, Wu WB. TNF-alpha induces CXCL1 chemokine expression and release in human vascular endothelial cells in vitro via two distinct signaling pathways. *Acta Pharmacol Sin* **35**, 339-350 (2014).
9. Cai JJ, Wen J, Jiang WH, Lin J, Hong Y, Zhu YS. Androgen actions on endothelium functions and cardiovascular diseases. *Journal of geriatric cardiology : JGC* **13**, 183-196 (2016).
10. Koryakina Y, Ta HQ, Gioeli D. Androgen receptor phosphorylation: biological context and functional consequences. *Endocrine-related cancer* **21**, T131-145 (2014).
11. Kim TK, Park JH. More about the basic assumptions of t-test: normality and sample size. *Korean journal of anesthesiology* **72**, 331-335 (2019).

12. Elliott AC, Woodward WA. *Statistical Analysis Quick Reference Guidebook: With SPSS Examples*. Sage Publications Pvt. Ltd. (2006).
13. Neville JA, Lang W, Fleischer AB, Jr. Errors in the Archives of Dermatology and the Journal of the American Academy of Dermatology from January through December 2003. *Archives of dermatology* **142**, 737-740 (2006).
14. Rochon J, Gondan M, Kieser M. To test or not to test: Preliminary assessment of normality when comparing two independent samples. *BMC medical research methodology* **12**, 81 (2012).